# CorresNeRF: Image Correspondence Priors for Neural Radiance Fields

**Yixing Lao**
The University of Hong Kong
yxlao@cs.hku.hk

**Xiaogang Xu**
Zhejiang Lab, Zhejiang University
xgxu@zhejianglab.com

**Zhipeng Cai**
Intel Labs
zhipeng.cai@intel.com

**Xihui Liu**
The University of Hong Kong
xihuiliu@eee.hku.hk

**Hengshuang Zhao**[*]
The University of Hong Kong
hszhao@cs.hku.hk

## Abstract

Neural Radiance Fields (NeRFs) have achieved impressive results in novel view synthesis and surface reconstruction tasks. However, their performance suffers under challenging scenarios with sparse input views. We present CorresNeRF, a novel method that leverages image correspondence priors computed by off-the-shelf methods to supervise NeRF training. We design adaptive processes for augmentation and filtering to generate dense and high-quality correspondences. The correspondences are then used to regularize NeRF training via the correspondence pixel reprojection and depth loss terms. We evaluate our methods on novel view synthesis and surface reconstruction tasks with density-based and SDF-based NeRF models on different datasets. Our method outperforms previous methods in both photometric and geometric metrics. We show that this simple yet effective technique of using correspondence priors can be applied as a plug-and-play module across different NeRF variants. The project page is at https://yxlao.github.io/corres-nerf/.

## 1   Introduction

Building on coordinate-based implicit representations [1, 2, 3], Neural Radiance Field (NeRF) [4] has achieved great success in solving the fundamental computer vision problem of reconstructing 3D geometries from RGB images, benefiting various downstream applications. However, training such implicit representations typically requires a large number of input views, especially for objects with complex shapes, which can be costly to collect. Therefore, training NeRFs with sparse input RGB views remains a challenging yet important problem, where solving it can benefit various real-world applications, e.g., 3D portrait reconstruction in the monitoring system [5, 6, 7], the city digital reconstruction [8, 9, 10], etc.

Several works [11, 12, 13, 14, 15, 16] have been proposed to address this problem by optimizing the rendering process or adding training constraints. However, these methods may suffer from poor real-world performance [12], since only sparse 2D input views are an under-constrained problem [17, 13], and the training process is prone to overfitting on the limited input views. Recent works have proposed to utilize extra priors to supervise NeRF training [14, 16, 18]. For example, some work proposed to train a separate network to compute depth priors [14]. However, current priors are not robust enough against the sparse property of the target scene, e.g., DS-NeRF [13] relies on running external SfM module [19] which may not yield sufficient point clouds for supervision, or not have an absolute scale and shift from the monocular depth estimation [16].

---

[*]Corresponding author

37th Conference on Neural Information Processing Systems (NeurIPS 2023).

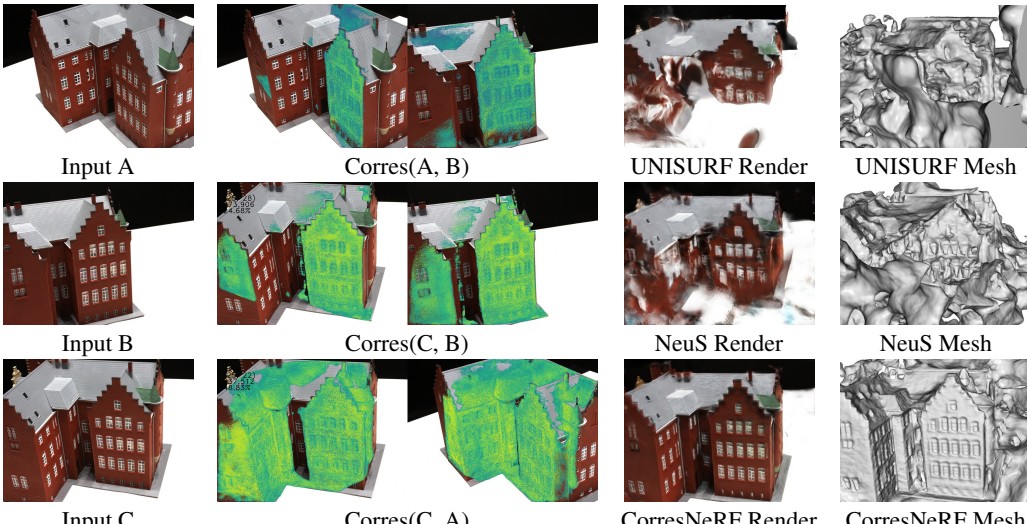

| Input A | Corres(A, B) | UNISURF Render | UNISURF Mesh |
| Input B | Corres(C, B) | NeuS Render | NeuS Mesh |
| Input C | Corres(C, A) | CorresNeRF Render | CorresNeRF Mesh |

Figure 1: **Novel view synthesis and surface reconstruction from sparse inputs with image correspondence priors**. Given a sparse set of input images (column 1), our method leverages the image correspondence priors computed from pre-trained models (column 2) to supervise NeRF training. The color of the highlighted pixels represents the confidence of the correspondence, where higher confidence values are greener. With image correspondence supervision, we achieve much higher quality novel view synthesis (column 3) and surface reconstruction (column 4) compared to the baseline UNISURF [20] and NeuS [21] models. The correspondence priors can be used in both density-based and SDF-based neural implicit representations on both novel-view synthesis and surface reconstruction tasks. Best viewed in color.

Our key observation is that image correspondences can provide strong supervision signals for the training of NeRF. In Figure 2, we visualize the point cloud created from triangulating the image correspondences pre-processed by our automatic augmentation and filtering (Section 3.2). Without training any NeRF networks, this triangulated point cloud clearly captures rich geometrical information about the target scene, which suggests that the image correspondences, along with the given camera parameters, can provide strong supervision signals for NeRF Training. The idea of using image correspondences as priors is widely applicable, since the correspondences can be estimated as long as there are sufficient textured overlapping regions among the input views, regardless of the NeRF variants. Additionally, the acquisition of image correspondence is inexpensive, as they can be directly computed using pre-trained off-the-shelf methods [22, 23, 24].

To improve the robustness of our method, we propose an automatic augmentation and outlier filtering process for the correspondences, ensuring their quantity and quality. Ablation studies demonstrate the effectiveness of our proposed augmentation and filtering strategy. To incorporate the correspondences into the NeRF training, we design novel correspondence loss terms, including pixel reprojection loss and depth loss. The reprojection loss is designed to constrain the distances between the reprojected corresponding points in the 2D pixel coordinates and the pixel-level distances. The depth loss is designed to constrain the relative depth differences between the corresponding points. Moreover, the confidence values from correspondence estimation are adopted as loss weights to avoid the negative impact of mismatching correspondences.

We evaluate our method on novel view synthesis and surface reconstruction tasks on LLFF [25] and DTU [26] datasets. We observe that the combination of correspondence priors via our method leads to significantly improved performance in novel-view synthesis (e.g., the PSNR of NeRF baseline [4] on LLFF is improved more than 3dB, SSIM is enhanced by 14%) and surface reconstruction (the DepthMAE of NeRF baseline on LLFF is decreased from 1.66 to 0.91, and Chamfer-$L_1$ distance is reduced from 6.16 to 2.63 for NeuS [21] on DTU). Furthermore, we benchmark our method against other state-of-the-art sparse-view reconstruction methods on different datasets, and show that our approach (built based on the simple typical NeRF, i.e., vanilla NeRF [4] for view synthesis and NeuS [21] for surface reconstruction) outperforms the comparative methods in terms of photometric and geometrical metrics.

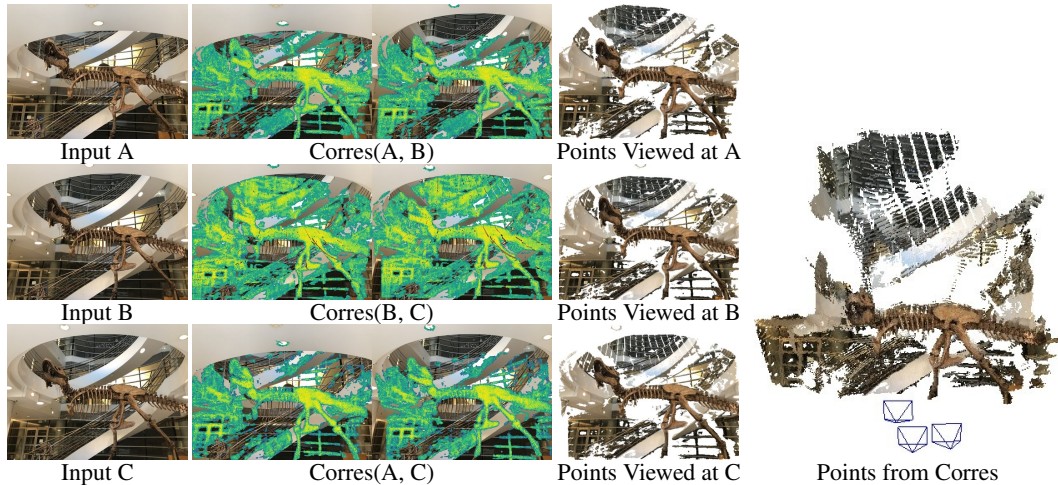

| Input A | Corres(A, B) | Points Viewed at A | |
| Input B | Corres(B, C) | Points Viewed at B | |
| Input C | Corres(A, C) | Points Viewed at C | Points from Corres |

Figure 2: **Reconstructing point cloud by triangulating dense image correspondences.** Column 1 shows the input images. Column 2 shows the correspondences between image pairs. With these correspondences and camera parameters, one can directly reconstruct 3D points without any training. Column 3 shows the reconstructed point cloud visualized using the same input cameras. The rightmost column shows the points rendered in novel view, as well as the camera poses. This example showcases the strong supervision signals that correspondences can bring to NeRF training. Best viewed in color.

We summarize our contributions below:

- We propose to use image correspondence as priors to supervise NeRF training, leading to better performance under challenging sparse-view scenarios.
- We design an adaptive pipeline to automatically augment and filter image correspondences, ensuring their quantity and quality.
- We propose robust correspondence loss, including pixel reprojection loss and depth loss based on correspondence priors.
- Extensive experiments are conducted on different baselines and datasets, showing the effectiveness of our method under different choices of neural implicit representations.

## 2 Related Work

**Nerual implicit representations.** Unlike traditional explicit 3D representations such as point cloud, mesh, or voxels, neural implicit representations [2, 1, 3] define a 3D scene with an implicit function that maps 3D coordinates and view directions to the corresponding properties (e.g., color and density) or features. The implicit function is typically modeled with neural networks. These types of representations are more compact and flexible towards different scenes, and they have been successfully applied to various tasks such as 3D reconstruction [4] and novel view synthesis [21].

**Sparse-view NeRFs.** Due to the practical limitation and the high cost of data collection, the data used for neural implicit representation learning is often sparse. To address this problem, researchers have recently proposed to incorporate geometric [27, 14, 13, 15, 28, 29, 16, 30] or pre-trained priors [12, 11, 31, 32, 33] to supervise the training of neural implicit representations. As a fundamental property of a scene, depth priors are commonly used, including depth generated from an SfM system [13], depth computed via monocular depth estimation [16], and depth predicted from a depth completion network [27, 14]. In contrast to these works, we rely on a more robust and flexible low-level feature, i.e., image correspondences, as supervision. The image correspondences obtained from modern image matchers [24, 23] are typically denser than the depth maps generated from traditional SfM system [19], when given sparse input views, since the matching is completed according to various characteristics in addition to geometrical information. With the known camera parameters, one can also compute absolute depth from the image correspondences, avoiding the ambiguity of shift and scale of depth maps from monocular depth estimation methods [16]. ConsistentNeRF [34] uses the depths from pre-trained MVSNeRF [31] to derive the correspondence mask to emphasize the multi-view appearance consistency, while our method uses the correspondences computed from off-the-shelf image matchers to regularize geometric properties of the scene. Concurrent work

SPARF [35] uses image correspondences to refine noisy poses from sparse input views with a 2D pixel-space correspondence loss, while our method uses both a 2D pixel-space loss and a 3D depth-space loss. Our depth-space loss is normalized to compute the relative depth differences, adapting to different scene scalings and geometry properties.

**Geometric matching.** The task of finding pixel-level correspondences between two views of a 3D scene, known as geometric matching, is a fundamental computer vision problem. Early works of geometric matching are based on matching measurement via hand-crafted local features [36, 37, 38]. Later, detectors and feature descriptors learned through data-driven processes [39, 40, 41, 22] were proposed to substitute the hand-crafted features, surpassing their performance. Recently, detector free methods [23], transformer-based [42, 43], and dense geometric matching [44, 45, 24] have been proposed to further improve the performance. In this work, we use image correspondences generated by the state-of-the-art dense image matcher [24] as supervision for neural implicit representations.

## 3 Method

### 3.1 Background of Neural Radiance Fields

For a given 3D point $\mathbf{x} \in \mathbb{R}^3$ and a viewing direction $\mathbf{d} \in \mathbb{R}^3$, a neural radiance field [4] predicts the corresponding density $\sigma \in [0, \infty)$ and RGB color $\mathbf{c} \in [0, 1]^3$, modeled by an MLP network, as

$$f_\theta : (\gamma(\mathbf{x}), \gamma(\mathbf{d})) \mapsto (\mathbf{c}, \sigma), \tag{1}$$

where $\gamma$ is the positional encoding function.

A ray is defined as $\mathbf{r}(t) = \mathbf{o} + t\mathbf{d}, t \in [t_n, t_f]$, where $\mathbf{o}$ is the camera center and $\mathbf{d}$ is the ray direction, $t_n$ is the near bound and $t_f$ is the far bound. To render the ray $\mathbf{r}$ with a pre-defined $t_n$ and $t_f$, we integrate the density $\sigma$ and color $\mathbf{c}$ along the ray, as

$$\hat{\mathbf{c}}_\theta(\mathbf{r}) = \int_{t_n}^{t_f} T(t)\sigma_\theta(\mathbf{r}(t))\mathbf{c}_\theta(\mathbf{r}(t), \mathbf{d})dt, \quad T(t) = \exp\left(-\int_{t_n}^{t} \sigma_\theta(\mathbf{r}(t))dt\right), \tag{2}$$

where $T(t)$ is the accumulated transmittance, $\mathbf{c}_\theta(\mathbf{r}(t), \mathbf{d})$ and $\sigma_\theta(\mathbf{r}(t))$ are the predicted color and density output from $f_\theta$, respectively. The rendering is implemented via stratified sampling approach, where $M$ points are sampled in $[t_n, t_f]$, as $\{x_1, ..., x_M\}$. The density and color can be obtained as

$$\hat{\mathbf{c}}_\theta(\mathbf{r}) = \sum_{i=1}^{M} T_i(1 - \exp(-\sigma_\theta(x_i)\delta_i))\mathbf{c}_\theta(x_i, \mathbf{d}), \quad T_i = \exp(-\sum_{j=1}^{i-1} \sigma_\theta(x_j)\delta_j), \tag{3}$$

where $\delta_j = t_{j+1} - t_j$ is the distance between adjacent samples. Specifically, for a ray $r$, its predicted 3D point can be obtained by summing up the weighted depth values along the ray, as

$$\mathbf{y} = \mathbf{o} + \left(\sum_{i=1}^{M} T_i(1 - \exp(-\sigma_\theta(x_i)\delta_i))t_i\right)\mathbf{d}. \tag{4}$$

To optimize parameter $\theta$ in the NeRF model, a set of input images and camera parameters are provided, and the mean squared error color loss is minimized for optimization, as

$$\mathcal{L}_{\text{color}}(\theta, \mathcal{R}) = \mathbb{E}_{\mathbf{r} \in \mathcal{R}} \|\hat{\mathbf{c}}_\theta(\mathbf{r}) - \mathbf{c}(\mathbf{r})\|_2^2, \tag{5}$$

where $\mathcal{R}$ is the set of rays in the training views, and $\mathbf{c}(\mathbf{r})$ is the ground-truth color of the ray $\mathbf{r}$.

### 3.2 Generating Correspondences

In this paper, we focus on how to utilize the computed image correspondences to enhance the performance of neural implicit representations in NeRF. Thus, the quality of correspondence is crucial. For each pair of images in the training views, we compute the correspondences using an off-the-shelf SOTA pre-trained image-matching model. In particular, DKMv3 [24] is used because it provides dense matching results, which is suitable for our use case. To improve generalization ability, we fuse the predictions of the indoor and outdoor models, which are pre-trained on ScanNet [46] and MegaDepth [47] respectively. To further enhance the reliability of the correspondences, we propose

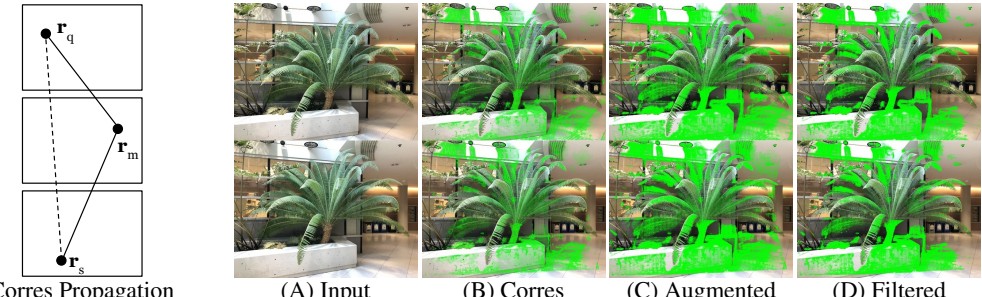

| Corres Propagation | (A) Input | (B) Corres | (C) Augmented | (D) Filtered |

Figure 3: **Correspondence generation process**. The left column illustrates correspondence propagation in a connected component in the correspondence graph $\mathcal{G}$. Given correspondence pairs $(\mathbf{r}_q, \mathbf{r}_m)$ and $(\mathbf{r}_m, \mathbf{r}_s)$ with confidence $\alpha_{q,m}$ and $\alpha_{m,r}$, we assign correspondence relationship to pair $(\mathbf{r}_q, \mathbf{r}_s)$ with $\alpha_{q,s} = \alpha_{q,m}\alpha_{m,s}$. The right columns show the overall correspondence generation process: (A) input images; (B) correspondence pairs generated by the vanilla matcher; (C) correspondence pairs augmented by image transformations and correspondence propagation; and (D) correspondence pairs after outlier removal. Best viewed in color.

to utilize the correspondence confidence, and also design the automatic and adaptive correspondence process algorithm, increasing the convincing correspondences and removing the outliers.

**Confidence.** For ray $\mathbf{r}_q$ in training rays $\mathcal{R}$, the image matcher computes a set of corresponding rays $\mathcal{C}(\mathbf{r}_q)$. For each pair of correspondences $\mathbf{r}_q$ (query) and $\mathbf{r}_s$ (support) in $\{(\mathbf{r}_q, \mathbf{r}_s) \mid \mathbf{r}_q \in \mathcal{R}, \mathbf{r}_s \in \mathcal{C}(\mathbf{r}_q)\}$, a confidence value $\alpha_{q,s} \in [0.5, 1]$ is also predicted by DKMv3. Note that a ray $\mathbf{r}_q$ can have 0, 1, or more corresponding rays $\mathbf{r}_s$ in different images. Confidence scores are used as the scaling factors for the correspondence losses, which will be described in Sec. 3.3.

**Augmentation.** To increase the number of correspondences, we perform augmentations for correspondences. The first type of augmentation is image transformations, including flipping, swapping query and support images, and scaling. These image transformations can effectively increase the density of the predicted correspondences since the image transformations can provide various context conditions to generate correspondences. The second type of augmentation propagates correspondences across image pairs, effectively increasing the area coverage of the correspondences. We build an undirected graph $\mathcal{G} = (\mathcal{V}, \mathcal{E})$, with vertices $\mathcal{V} = \{\mathbf{r} \mid \mathbf{r} \in \mathcal{R}\}$, and edges $\mathcal{E} = \{(\mathbf{r}_q, \mathbf{r}_s) \mid \mathbf{r}_s \in \mathcal{C}(\mathbf{r}_q)\}$. For each edge $(\mathbf{r}_q, \mathbf{r}_s)$, a confidence value $\alpha_{q,s}$ is assigned. We then propagate the correspondence relationship to vertex pairs within each connected component in $\mathcal{G}$. In particular, let $\mathbf{r}_q$ and $\mathbf{r}_s$ be two vertices with distance $d$, where is a path $(\mathbf{r}_q, \mathbf{r}_1, \mathbf{r}_2, \ldots, \mathbf{r}_{d-1}, \mathbf{r}_s)$ connecting them. We assign a correspondence relationship between $\mathbf{r}_q$ and $\mathbf{r}_s$ with confidence $\alpha_{q,s} = \alpha_{q,1}\alpha_{1,2}\ldots\alpha_{d-1,s}$. In practice, we cap the propagation distance $d \leq d_{\max}$, where we use $d_{\max} = 2$ in our experiments. Figure 3 (B) and (C) show the original and augmented correspondences, respectively.

**Outlier filtering.** To increase the quality of the correspondences to guide the supervision, we remove outliers after calculating and enhancing the correspondences. First, we remove outliers according to the projected ray distance between corresponding points. Suppose $\mathbf{p}_q$ and $\mathbf{p}_s$ be a pair of 2D correspondence in $I_q$ to $I_s$, $\pi_q$ and $\pi_s$ be the world-to-pixel projection for $I_q$ and $I_s$, respectively. Given a pair of correspondences and camera parameters, we compute the closest 3D points $\mathbf{x}_q$ and $\mathbf{x}_s$ along the two rays shot from the camera centers through the two correspondences. Then, we project these two 3D points to the correspondence's image plane. The projected ray distance [48] is defined as the averaged Euclidean distance between the projected points and the correspondences:

$$d_{\mathrm{proj}} = \frac{\|\pi_q(\mathbf{x}_s) - \mathbf{p}_q\|_2 + \|\pi_s(\mathbf{x}_q) - \mathbf{p}_s\|_2}{2}. \tag{6}$$

We remove the correspondences with projected ray distance $d_{\mathrm{proj}}$ larger than a threshold. Second, we remove outliers by checking if a point is statistically far from its neighbors. For each pair of correspondences, two 3D points $\mathbf{x}_q$ and $\mathbf{x}_s$ can be obtained, which has been indicated in the last paragraph. We then consider $\frac{1}{2}(\mathbf{x}_q + \mathbf{x}_s)$ to be the 3D point of the correspondence. We do this for all correspondence pairs to obtain a set of 3D points $\mathcal{P}$. For each 3D point in $\mathcal{P}$, we compute the average distance to its $k$ nearest neighbors and remove the point (as well as its matched correspondence pair) if the distance is larger than a threshold. This threshold is determined by the standard deviation of all points' average distances in $\mathcal{P}$.

Experiments in Sec. 4.5 demonstrate the effects of our proposed augmentation and removal strategy since they can increase the reliability of the correspondences utilized in NeRF's training.

### 3.3 Correspondence Loss

We incorporate the obtained correspondences into NeRF training from two aspects: correspondence pixel reprojection loss and depth loss, as illustrated in Fig. 4.

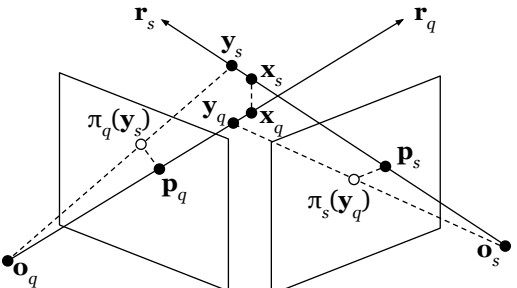

Figure 4: **Correspondence loss of a ray pair** $(\mathbf{r}_q, \mathbf{r}_s)$. $\mathbf{o}_q$ and $\mathbf{o}_s$ are the ray origins. $\mathbf{x}_q$ and $\mathbf{x}_s$ are the closest points along the two rays. $\mathbf{y}_q$ and $\mathbf{y}_s$ are the 3D points predicted by the NeRF model. $\pi_q(\mathbf{y}_s)$ and $\pi_s(\mathbf{y}_q)$ are the 2D projections of $\mathbf{y}_s$ and $\mathbf{y}_q$ on the other image. The correspondence losses are defined with the geometric properties as illustrated.

**Correspondence pixel reprojection loss.** Let $\mathbf{y}_q$ and $\mathbf{y}_s$ be the corresponding 3D points predicted via NeRF, by accumulating the weighted sum of z-depth values to the ray origin according to Eq. 4. We compute the pixel-space reprojection loss as follows:

$$\mathcal{L}_{\text{pixel}}(\theta, \mathcal{R}) = \mathbb{E}_{\mathbf{r}_q \in \mathcal{R}, \mathbf{r}_s \in \mathcal{C}(\mathbf{r}_q)} \, \alpha_{q,s} \left( \|\pi_q(\mathbf{y}_s) - \mathbf{p}_q\|_2 + \|\pi_s(\mathbf{y}_q) - \mathbf{p}_s\|_2 \right). \tag{7}$$

The pixel-space reprojection loss is similar to the projected ray distance [48]. The difference is that we reproject the predicted 3D points $\mathbf{y}_q$ and $\mathbf{y}_s$, instead of reprojecting the closest points along the two rays. Moreover, we utilize the correspondence score $\alpha_{q,s}$ as weights to provide a more reliable metric. Since the camera parameters are known, the projected ray distance is constant. However, the reprojection of the predicted 3D points can provide a supervision signal to guide the NeRF model.

**Correspondence depth loss.** The correspondence depth loss term penalizes the difference between the predicted depth and the ground-truth depth. The ground-truth depth is estimated by computing the closest 3D points $\mathbf{x}_q$ and $\mathbf{x}_s$ along the two rays shot from the camera centers $\mathbf{o}_q$ and $\mathbf{o}_s$ through the two correspondences. We use a relative depth loss, which is adaptive to different scalings and can be described with the following equation:

$$\mathcal{L}_{\text{depth}}(\theta, \mathcal{R}) = \mathbb{E}_{\mathbf{r}_q \in \mathcal{R}, \mathbf{r}_s \in \mathcal{C}(\mathbf{r}_q)} \, \alpha_{q,s} \left( \left| \frac{\|\mathbf{y}_q - \mathbf{o}_q\|_2}{\|\mathbf{x}_q - \mathbf{o}_q\|_2} - 1 \right| + \left| \frac{\|\mathbf{y}_s - \mathbf{o}_s\|_2}{\|\mathbf{x}_s - \mathbf{o}_s\|_2} - 1 \right| \right), \tag{8}$$

where $\alpha_{q,s}$ is the confidence of correspondence to regular the effects of the relative depth loss. In the ideal scenario, $\mathbf{y}_q$ should overlap with $\mathbf{x}_q$ and similarly $\mathbf{y}_s$ should overlap with $\mathbf{x}_s$, so the loss will be zero. The relative depth loss is a smooth function and is minimized when the predicted depth is close to the estimated depth of the ground truth.

**Training procedure.** For each ray in a batch of rays $\mathcal{R}_{\text{batch}}$, we query the correspondence map to find the corresponding rays in all training rays $\mathcal{R}$. We assume that each ray may have $B$ corresponding rays, $B \in [0, +\infty)$. For each pair of corresponding rays, we run the forward pass of the NeRF model to obtain the predicted 3D points. We then compute the pixel-space reprojection loss and the correspondence depth loss for each pair of corresponding rays. The final loss is the sum of pixel-space correspondence reprojection loss, correspondence depth loss, and the regular NeRF color loss $\mathcal{L}_{\text{color}}$:

$$\mathcal{L} = \mathcal{L}_{\text{color}} + \lambda_1 \mathcal{L}_{\text{pixel}} + \lambda_2 \mathcal{L}_{\text{depth}}, \tag{9}$$

where $\lambda_1$ and $\lambda_2$ are the loss weights, and we set them to $0.1$ in our experiments.

## 4 Experiments

### 4.1 Datasets

We compare novel view synthesis results on LLFF [25] with density-field-based NeRF models. We follow the convention to use every 8th image as test images [4], while selecting the training views uniformly from the rest of the images [29]. The selected training views and test views are the same across all methods. Three input views are used for training.

We compare surface reconstruction results on DTU [26] with SDF-based NeRF models. We follow the convention to use the same 15 scenes from DTU as previous works [21, 49]. The selected training views and test views are the same across all methods. Three input views are used for training.

Table 1: **Quantitative results on LLFF.** We compare novel view synthesis on the LLFF dataset for 3 input views. Our method outperforms the baseline NeRF model and the sparse-view optimized models. Our model is built with the vanilla NeRF as the direct baseline.

| | PSNR ↑ | SSIM ↑ | LPIPS ↓ | Depth MAE ↓ |
|---|---|---|---|---|
| NeRF [4] | 16.79 | 0.56 | 0.37 | 1.66 |
| DS-NeRF [13] | 17.09 | 0.57 | 0.38 | 1.64 |
| RegNeRF [29] | 19.08 | 0.59 | 0.34 | 1.02 |
| Ours | **19.83** | **0.70** | **0.29** | **0.91** |

Table 2: **Quantitative results on DTU.** We compare novel view synthesis and surface reconstruction with SDF-based methods on the DTU dataset for 3 input views. All models are trained without foreground masks. For evaluation, we report photometric metrics for the masked foreground object and the full image. The Chamfer-$L_1$ geometric metric is computed with the official DTU evaluation code with foreground masks applied. Our model is built with NeuS as the direct baseline, achieving the best results across photometric and geometric metrics.

| | PSNR ↑ | | SSIM ↑ | | LPIPS ↓ | | Chamfer-$L_1$ ↓ |
|---|---|---|---|---|---|---|---|
| | Object | Image | Object | Image | Object | Image | Object |
| UNISURF [20] | 12.90 | 13.85 | 0.47 | 0.58 | 0.28 | 0.57 | 7.23 |
| VolSDF [52] | 15.68 | 15.44 | 0.58 | 0.65 | 0.21 | 0.47 | 4.44 |
| NeuS [21] | 16.06 | 16.37 | 0.59 | 0.66 | 0.21 | 0.46 | 6.16 |
| Ours | **20.58** | **18.23** | **0.77** | **0.76** | **0.13** | **0.33** | **2.63** |

## 4.2 Evaluation Metrics

For the novel view synthesis task, we report photometric metrics, including PSNR, structural similarity index (SSIM) [50], and the learned perceptual metric LPIPS [51]. In addition to the image metrics, we also report the mean absolute error (MAE) for depth prediction. We compute the pseudo ground truth depth using the baseline NeRF model given *all* input views. The depth error is calculated as the mean absolute difference between the predicted depths and the pseudo ground truth depths. Both the predicted depths and the pseudo ground truth depths are in the normalized coordinates.

For the surface reconstruction task, we report the Chamfer distance with DTU's official evaluation code [26], where the Chamfer distance is computed in the unnormalized world coordinates for the foreground objects. We also report the PSNR, SSIM, and LPIPS metrics for the rendered images with (object) and without (image) the foreground masks.

## 4.3 Novel View Synthesis

**Quantitative comparison.** For the novel view synthesis task, we compare our method with the baseline NeRF [4] as well as few-view optimized DS-NeRF [13] and RegNeRF [29]. DS-NeRF uses external SfM module COLMAP [19] to generate sparse point clouds for depth supervision, while RegNeRF applies additional regularizations on depths and colors rendered from unobserved views. The experiments are performed on the LLFF dataset with 3 input views. Our method outperforms all other methods regarding photometric metrics and depth prediction. The results are shown in Table 1. For DS-NeRF, we observe that if COLMAP is only given sparse-view inputs with known camera poses for SfM (as opposed to providing all views for SfM and selecting visible points from sparse views), COLMAP does not generate a sufficient number of points for depth supervision. In contrast, our method relies on image correspondences as inputs that are computed according to various characteristics. Therefore, it can leverage much denser prior information for supervision, as shown in Fig. 3.

**Qualitative comparison.** Besides the quantitative comparison, we provide the qualitative comparison to demonstrate the effects of our approach in improving novel view synthesis performance. The visual samples are shown in Fig. 5. Compared with baselines, our results have sharper and more accurate details and fewer artifacts. Such an improvement in performance is most visually evident in challenging cases such as small or thin structures.

### 4.4 Surface Reconstruction

**Quantitative comparison.** We apply the same correspondence loss terms to the SDF-based neural implicit field method NeuS [21] and compare the results with the baseline NeuS model as well as other two SOTA SDF-based surface reconstruction method, UNISURF [20] and VolSDF [52]. The experiments are performed on the DTU dataset with 3 input views. The results are shown in Table 2. Our method outperforms all baseline models in terms of Chamfer-$L_1$ distances, i.e., ours generates more accurate surfaces. Moreover, ours also is also significantly better than the baselines in terms of rendering metrics, including PSNR, SSIM, and LPIPS. This further demonstrates that our approach can be plugged into different types of network backbones (density-based and SDF-based) and can be applied to different tasks (novel view synthesis and surface reconstruction).

**Qualitative comparison.** The visual comparisons between our method and the baseline for surface reconstruction are shown in Fig. 6. It's apparent that our 3D reconstruction results have more precise geometrical shapes. Combined with the correspondence priors, our CorresNeRF achieves clean and high-quality reconstruction even when given sparse input views.

### 4.5 Ablation Studies

We perform ablation studies of the correspondence generation process, correspondence loss terms, robustness to correspondence noise, and the effect of foreground masks.

**Correspondence generation process.** We compare the result with augmentation removed, with automatic filtering removed, and with the full pipeline. The results are shown in the first two rows in Table 3. The result shows that both the augmentation and the automatic outlier filtering process are able to improve the reconstruction quality measured in photometric and geometric metrics. The augmentation step can provide denser supervision signals, while the filtering step is able to remove the noisy correspondences to further improve the reconstruction quality.

**Correspondence loss terms.** We evaluate the effectiveness of the correspondence pixel reprojection loss and correspondence depth loss terms. We compare the results with only the reprojection loss removed, with only the depth loss removed, and the full pipeline. The results are displayed in Table 3. Our results show that both correspondence reprojection loss and the correspondence depth loss contribute significantly to the reconstruction quality, each adding ∼1 dB PSNR compared to the vanilla NeRF. The combined loss term yields the best performance, adding ∼3 dB PSNR.

**Robustness to correspondence noise.** To evaluate the robustness of CorresNeRF to noisy correspondences, we introduce Gaussian noise with standard deviations of 1, 2, and 4 pixels to both the $x$ and $y$ pixel coordinates. We measure the performance of CorresNeRF on LLFF dataset with 3 input views. The results are shown in Table 4. The automatic filtering process is able to remove the noisy correspondences, demonstrating the robustness of CorresNeRF to noisy correspondences.

**Correspondence with mask supervision.** For the surface reconstruction task, we evaluate the performance of CorresNeRF when the foreground mask is provided. When the foreground mask is available, the correspondences are filtered by the mask, allowing the model to focus on the foreground region. The results are shown in Table 5. Our model outperforms the NeuS model in both the with-mask and without-mask settings. Qualitative visualization results are shown in Fig. 8 in the supplementary material.

## 5 Conclusion

We presented CorresNeRF, a method that can utilize the image correspondence priors for training neural radiance fields with sparse-view inputs. We propose automatic augmentation and filtering methods to generate dense and high-quality image correspondences from sparse-view inputs. We design reprojection and depth loss terms based on the correspondence priors to regularize the neural radiance field training. Experiments show that our method can significantly improve the reconstruction quality measured in photometric and geometric metrics with only a few input images.

**Limitations and future work.** In CorresNeRF, convincing correspondences are obtained by using SOTA matching networks and further processed via adaptive augmentation and outlier removal. An

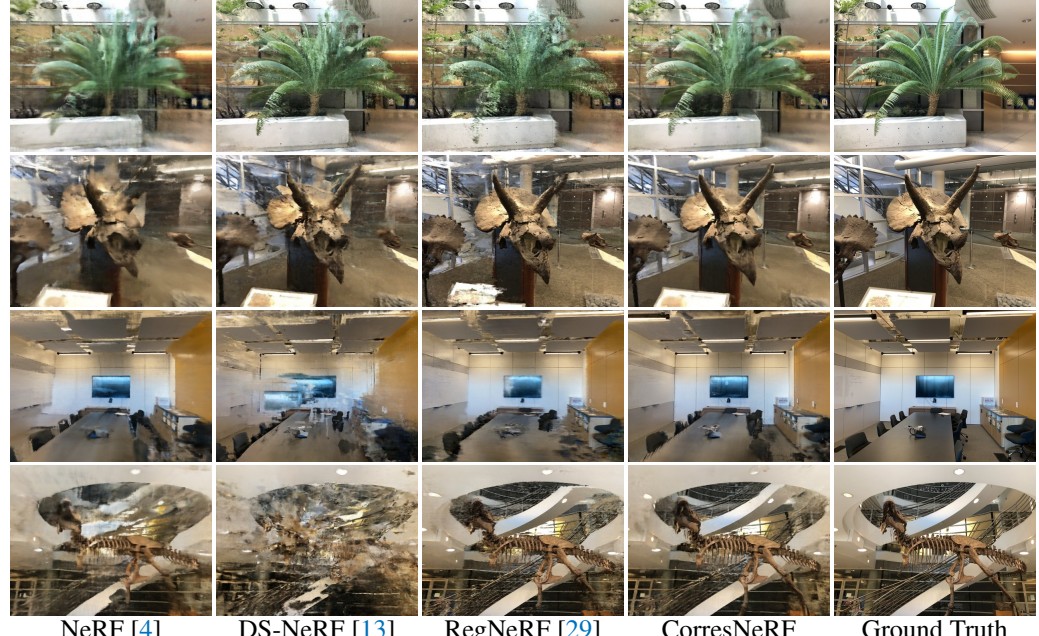

| NeRF [4] | DS-NeRF [13] | RegNeRF [29] | CorresNeRF | Ground Truth |
|---|---|---|---|---|

Figure 5: **Qualitative results from the LLFF dataset.**

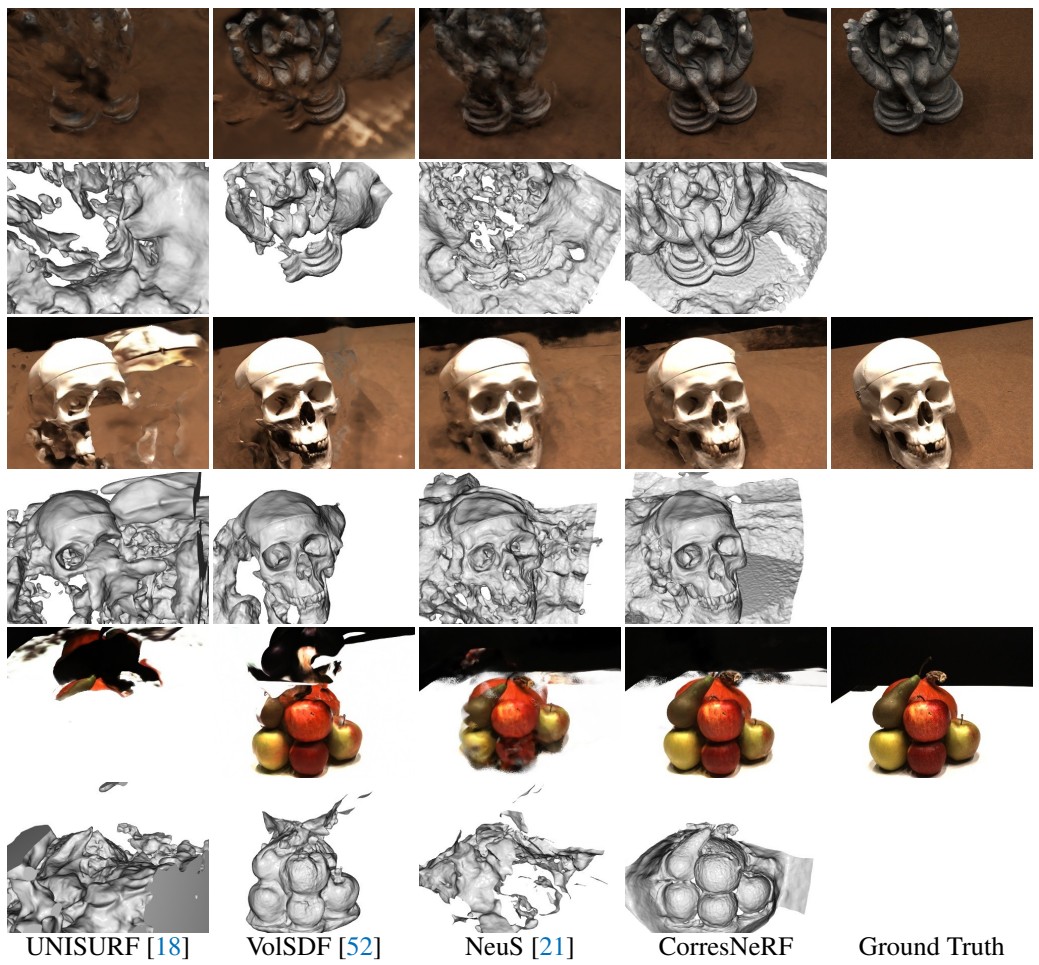

| UNISURF [18] | VolSDF [52] | NeuS [21] | CorresNeRF | Ground Truth |
|---|---|---|---|---|

Figure 6: **Qualitative results from the DTU dataset.**

Table 3: **Ablation study on correspondence generation and loss terms.** We verify the effectiveness of correspondence pre-processing and correspondence loss terms. We find that both correspondence augmentation and outlier filtering are helpful for the final performance. We also find that both the pixel reprojection loss and correspondence depth loss contribute to the final performance on both photometric and geometric metrics.

| | PSNR ↑ | SSIM ↑ | LPIPS ↓ | Depth MAE ↓ |
|---|---|---|---|---|
| NeRF [4] | 16.79 | 0.56 | 0.37 | 1.66 |
| Ours w/o Corres Augment | 18.50 | 0.66 | 0.32 | 1.15 |
| Ours w/o Corres Filter | 19.32 | 0.69 | 0.29 | 0.93 |
| Ours w/o Pixel Loss | 18.16 | 0.62 | 0.42 | 1.33 |
| Ours w/o Depth Loss | 17.75 | 0.59 | 0.51 | 1.47 |
| Ours | **19.83** | **0.70** | **0.29** | **0.91** |

Table 4: **Ablation study on robustness to correspondence noise.** We add Gaussian noise to the correspondences generated by our method and evaluate the performance of CorresNeRF on the LLFF dataset with 3 input views.

| | Corres % After Filter | PSNR ↑ | SSIM ↑ | LPIPS ↓ | Depth MAE ↓ |
|---|---|---|---|---|---|
| NeRF [4] | N/A | 16.79 | 0.56 | 0.37 | 1.66 |
| Ours (Noise STD = 4 px) | 13.96% | 18.31 | 0.61 | 0.48 | 1.04 |
| Ours (Noise STD = 2 px) | 27.04% | 19.16 | 0.66 | 0.33 | 1.06 |
| Ours (Noise STD = 1 px) | 48.91% | 19.31 | 0.67 | 0.28 | 1.06 |
| Ours (Noise STD = 0 px) | 100.00% | **19.83** | **0.70** | **0.29** | **0.91** |

Table 5: **Ablation study on the effects of foreground masks.** We report the results of NeuS and our model trained with and without foreground masks on the DTU dataset with 3 input views. The photometric metrics are calculated with the foreground masks applied regardless of the training settings.

| | PSNR ↑ | SSIM ↑ | LPIPS ↓ | Depth MAE ↓ |
|---|---|---|---|---|
| NeuS w/o Mask | 16.06 | 0.59 | 0.21 | 6.16 |
| Ours w/o Mask | **20.58** | **0.77** | **0.13** | **2.63** |
| NeuS w/ Mask | 20.85 | 0.78 | 0.12 | 2.36 |
| Ours w/ Mask | **21.85** | **0.81** | **0.11** | **1.57** |

admired number of correspondences can be acquired for most practical scenes (proved by experiments on various benchmarks). However, there are still some extreme cases where few correspondences are computed via matching networks (due to unreasonable camera positions or some specific textures of target scenes). Correspondence generation methods are needed for these cases to synthesize convincing correspondences along with the optimization of NeRF. Some recent works [53, 54] have proved that neural implicit representations can be utilized to learn correspondences in turn. Dealing with such extreme scenes will be our future work.

## Acknowledgment

This work is supported in part by the National Natural Science Foundation of China (No. 62201484), HKU Startup Fund, and HKU Seed Fund for Basic Research.

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

# Appendix

## A   Discussions on Correspondence

In this section, we provide additional discussions and visualizations on the correspondence generation pipeline, including direct correspondence reconstruction, comparing correspondence with SfM, and the correspondence propagation process.

### A.1   Comparing with SfM

We compare the DS-NeRF [13]'s SfM pipeline with our CorresNeRF's correspondence generation procedure in terms of the number of points generated and the number of correspondences generated. The results are reported in Table 6. The table shows that CorresNeRF is able to leverage much denser supervision compared to SfM-based depth supervision methods.

Table 6: **Comparison of DS-NeRF's SfM point cloud number, CorresNeRF's correspondence number, and their pixel coverage (%).** We compare the number of points in the supervision point cloud used in DS-NeRF [13] generated by COLMAP SfM [19] with the number of correspondences used in CorresNeRF. The numbers are reported in counts (total number of points and total number of correspondences) and pixel coverage percentage (the percentage of pixels that are supervised by sparse point cloud in DS-NeRF and the percentage of pixels having correspondences in CorresNeRF). For DS-NeRF, we run COLMAP only on the selected input views with camera poses provided (as opposed to providing all views for SfM and selecting visible points from input views). The results are reported on the LLFF dataset with 3 input views.

| | DS-NeRF [13] with SfM | Ours | | |
| --- | --- | --- | --- | --- |
| | | Base | Base+Filter | Base+Filter+Aug |
| fern | 362 (0.19%) | 325,633 (57%) | 281,382 (49%) | 368,798 (65%) |
| flower | 685 (0.35%) | 415,459 (73%) | 278,180 (49%) | 356,044 (62%) |
| fortress | 609 (0.31%) | 435,676 (76%) | 377,956 (66%) | 430,044 (75%) |
| horns | 512 (0.27%) | 270,778 (47%) | 224,812 (39%) | 271,705 (48%) |
| leaves | 201 (0.11%) | 272,158 (48%) | 153,024 (27%) | 198,412 (35%) |
| orchids | 229 (0.12%) | 213,660 (37%) | 153,250 (27%) | 242,620 (42%) |
| room | 345 (0.18%) | 226,156 (40%) | 190,302 (33%) | 260,308 (46%) |
| trex | 644 (0.34%) | 202,650 (35%) | 164,047 (29%) | 233,950 (41%) |

### A.2   Correspondence Propagation

We provide additional visualizations to demonstrate the correspondence propagation process. With correspondence propagation, image pairs initially with few or no correspondences may still obtain correspondence relationships propagated by other image pairs. For the propagated correspondences, we assign a new confidence score as the cumulative product of the confidence scores in the propagation path. These confidence levels are used to weigh the loss terms, as described in the main paper. See Fig. 7 for the visualizations.

## B   Implementation Details

**LLFF dataset.**   For the novel view synthesis task on LLFF, we based our code on NeRF [4]. For most parameters, we use the same settings as the original NeRF paper.

- Number of input views: 3
- Image scale factor: 8
- Correspondence pixel reprojection weight ($\mathcal{L}_{\text{pixel}}$): 0.1
- Correspondence depth weight ($\mathcal{L}_{\text{depth}}$): 0.1
- Training iterations: 50K[2]
- Learning rate: $5e^{-4}$

---

[2]Similar to RegNeRF [29]'s 44K iterations in a 3-view setting. More iterations could enhance results.

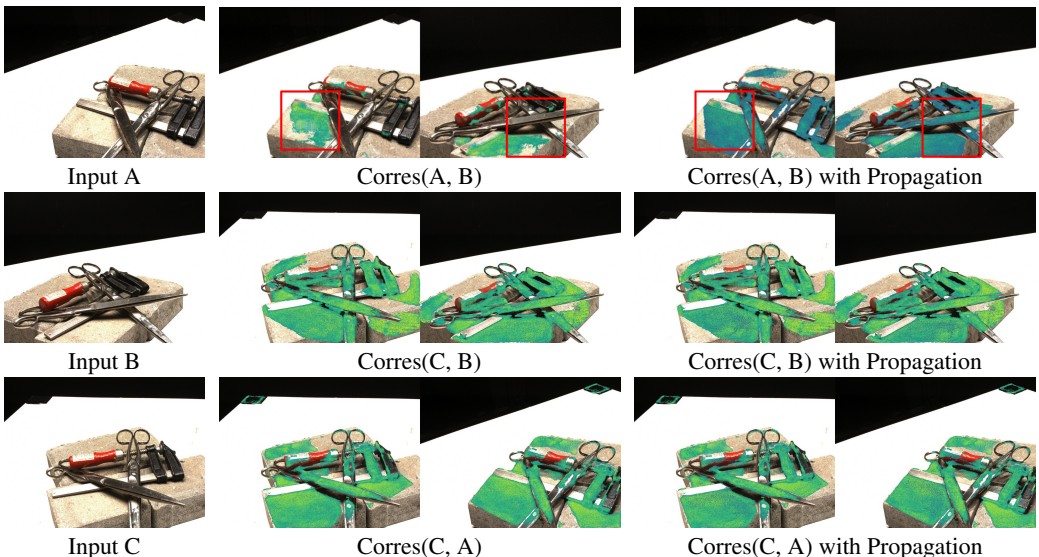

Figure 7: **Correspondence propagation visualizations**. Column 1 shows the sparse set of input images. Column 2 shows the correspondences without correspondence propagation but with all other augmentations and filtering enabled. Column 3 shows the propagated correspondences. The correspondence colors represent the confidence scores, where the higher confidence scores are greener. The highlighted area shows that the image pair (A, B) can receive additional correspondence relationships by propagating their correspondences from C.

**DTU dataset.** For the novel view synthesis and surface reconstruction task on DTU, we based our code on NeuS [21]. Important parameters are listed below. For most parameters, we utilize the same settings as in the original NeuS paper.

- Number of input views: 3
- Image scale factor: 4
- Correspondence pixel reprojection weight ($\mathcal{L}_{\text{pixel}}$): 0.1
- Correspondence depth weight ($\mathcal{L}_{\text{depth}}$): 0.1
- Training iterations: 200K
- Learning rate: $5e^{-4}$

# C  Additional Results

## C.1  Effects of Foreground Masks

We provide additional qualitative results to demonstrate the effects of foreground masks with correspondence priors in Fig. 8.

## C.2  Per-Scene Results

We provide per-scene results for the novel view synthesis task on the LLFF dataset in Table 7, as well as novel view synthesis and surface reconstruction tasks on the DTU dataset in Table 8 and Table 9.

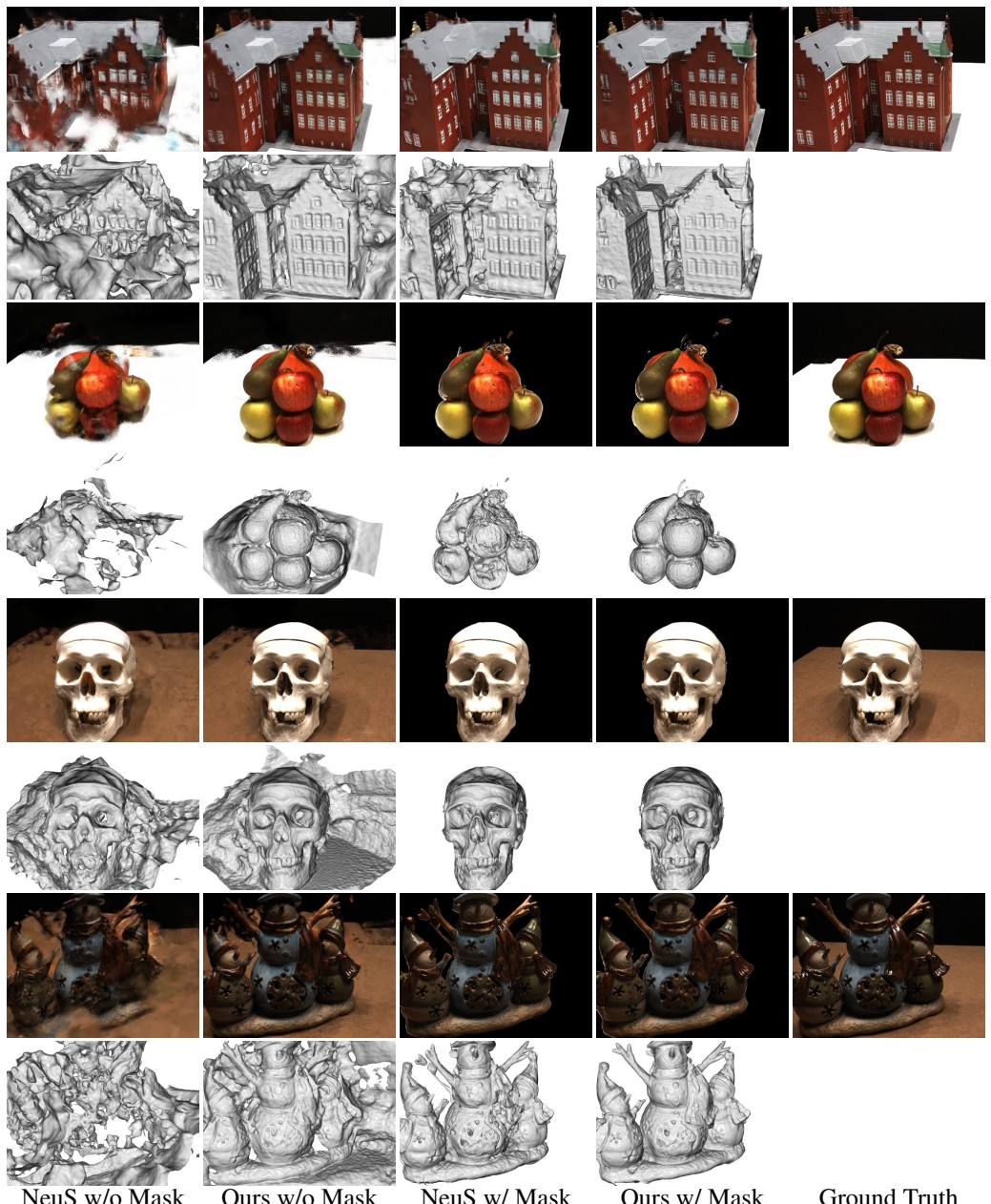

NeuS w/o Mask     Ours w/o Mask     NeuS w/ Mask     Ours w/ Mask     Ground Truth

Figure 8: **Effects of foreground masks.** Columns 1 and 2 compare NeuS [21] and our model trained without foreground masks. Our model outperforms NeuS in terms of novel-view rendering quality and mesh reconstruction quality. Columns 3 and 4 compare NeuS and our model trained with foreground masks. Our model trained with foreground masks further improves the rendering quality and geometric reconstruction quality. We observe more geometric details and fewer artifacts in our reconstruction compared with NeuS.

Table 7: **Per-scene results on the LLFF dataset.** We compare novel view synthesis on the LLFF dataset for 3 input views. Our method outperforms the baseline NeRF model and the sparse-view optimized DS-NeRF and RegNeRF models. Our model is built with the vanilla NeRF as the direct baseline. Note that for RegNeRF [29], the numbers reported here are results from our own evaluation, which are slightly better than the numbers (PSNR: 19.08, SSIM: 0.59, and LPIPS: 0.34) reported in the original author's paper and Table 1.

| | NeRF [4] / DS-NeRF [13] / RegNeRF [29] / Ours | | | |
| | PSNR ↑ | SSIM ↑ | LPIPS ↓ | Depth MAE ↓ |
|---|---|---|---|---|
| fern | 20.47 / **21.28** / 19.87 / 21.15 | 0.71 / **0.75** / 0.70 / 0.74 | 0.33 / **0.27** / 0.30 / 0.33 | **0.28** / 0.29 / 0.34 / 0.31 |
| flower | 17.35 / 19.35 / 19.93 / **20.44** | 0.56 / 0.67 / 0.68 / **0.69** | 0.34 / 0.26 / **0.23** / 0.25 | 1.78 / 0.85 / 0.91 / **0.85** |
| fortress | 19.48 / 20.87 / **23.33** / 22.53 | 0.68 / 0.67 / 0.74 / **0.76** | 0.38 / 0.37 / **0.26** / 0.35 | 0.95 / 0.67 / **0.46** / 0.51 |
| horns | 14.46 / 14.50 / 15.64 / **19.18** | 0.49 / 0.48 / 0.61 / **0.71** | 0.51 / 0.49 / 0.36 / **0.32** | 3.06 / 2.92 / 1.68 / **1.09** |
| leaves | 12.73 / 14.83 / 16.60 / **16.89** | 0.28 / 0.43 / **0.61** / 0.59 | 0.45 / 0.41 / **0.22** / 0.32 | 4.16 / **2.91** / 3.08 / 3.10 |
| orchids | 13.92 / 14.44 / 15.56 / **15.85** | 0.38 / 0.41 / **0.50** / 0.48 | 0.33 / 0.32 / **0.25** / 0.38 | 1.13 / 0.96 / 0.71 / **0.56** |
| room | 19.97 / 17.69 / 21.53 / **22.41** | 0.82 / 0.71 / 0.86 / **0.87** | 0.25 / 0.40 / **0.18** / 0.19 | 0.77 / 2.14 / 0.45 / **0.38** |
| trex | 15.89 / 13.79 / 20.16 / **20.18** | 0.59 / 0.43 / **0.77** / 0.76 | 0.36 / 0.50 / **0.20** / 0.22 | 1.19 / 2.39 / **0.51** / 0.51 |
| mean | 16.79 / 17.09 / 19.08 / **19.83** | 0.56 / 0.57 / 0.68 / **0.70** | 0.37 / 0.38 / **0.25** / 0.29 | 1.66 / 1.64 / 1.02 / **0.91** |

Table 8: **Per-scene results on the DTU dataset (image-level eval).** We compare novel view synthesis and surface reconstruction on the DTU dataset for 3 input views. The models are trained without mask supervision. The models are evaluated without foreground masks applied (image-level).

| | UNISURF [20] / VolSDF [52] / NeuS [21] / Ours | | |
| | PSNR ↑ | SSIM ↑ | LPIPS ↓ |
|---|---|---|---|
| scan24 | 8.98 / 10.53 / 10.36 / **13.99** | 0.43 / 0.46 / 0.46 / **0.64** | 0.65 / 0.58 / 0.61 / **0.36** |
| scan37 | 7.26 / 10.40 / 10.31 / **13.38** | 0.38 / 0.57 / 0.47 / **0.59** | 0.67 / 0.43 / 0.58 / **0.40** |
| scan40 | 12.14 / 7.48 / 10.61 / **13.51** | 0.58 / 0.39 / 0.48 / **0.70** | 0.59 / 0.67 / 0.63 / **0.38** |
| scan55 | 14.68 / 17.51 / 17.75 / **18.99** | 0.52 / 0.68 / 0.62 / **0.73** | 0.71 / 0.45 / 0.52 / **0.41** |
| scan63 | 4.79 / 5.80 / 10.92 / **12.66** | 0.42 / 0.53 / 0.65 / **0.78** | 0.75 / 0.61 / 0.50 / **0.31** |
| scan65 | 12.78 / 19.93 / 19.15 / **20.13** | 0.60 / **0.82** / 0.76 / 0.79 | 0.59 / 0.32 / 0.37 / **0.30** |
| scan69 | 17.05 / 16.62 / 17.87 / **19.71** | 0.57 / 0.62 / 0.62 / **0.75** | 0.51 / 0.46 / 0.48 / **0.31** |
| scan83 | 8.37 / 13.19 / 13.73 / **13.98** | 0.48 / 0.68 / 0.69 / **0.74** | 0.68 / 0.49 / 0.47 / **0.39** |
| scan97 | 13.28 / 10.93 / 14.57 / **15.80** | 0.59 / 0.54 / 0.67 / **0.74** | 0.46 / 0.51 / 0.39 / **0.31** |
| scan105 | 11.78 / 13.69 / 12.91 / **14.68** | 0.57 / 0.63 / 0.60 / **0.72** | 0.56 / 0.49 / 0.51 / **0.37** |
| scan106 | 18.10 / 21.36 / 20.09 / **22.03** | 0.71 / 0.78 / 0.76 / **0.83** | 0.44 / 0.35 / 0.40 / **0.30** |
| scan110 | 18.07 / 16.21 / 20.51 / **22.05** | 0.67 / 0.60 / 0.77 / **0.81** | 0.48 / 0.62 / 0.39 / **0.34** |
| scan114 | 20.90 / 23.00 / 23.12 / **24.25** | 0.79 / 0.85 / 0.83 / **0.86** | 0.37 / 0.26 / 0.28 / **0.24** |
| scan118 | 18.84 / 20.93 / 20.55 / **25.01** | 0.66 / 0.78 / 0.73 / **0.85** | 0.57 / 0.46 / 0.44 / **0.27** |
| scan122 | 20.69 / **24.02** / 23.08 / 23.26 | 0.70 / **0.85** / 0.80 / 0.81 | 0.53 / **0.32** / 0.38 / 0.33 |
| mean | 13.85 / 15.44 / 16.37 / **18.23** | 0.58 / 0.65 / 0.66 / **0.76** | 0.57 / 0.47 / 0.46 / **0.33** |

Table 9: **Per-scene results on the DTU dataset (object-level eval).** We compare novel view synthesis and surface reconstruction on the DTU dataset for 3 input views. Note that VolSDF fails to reconstruct a mesh for one of the scenes (scan110), thus its Chamfer-$L_1$ is averaged over the remaining scenes. The models are trained without mask supervision. The models are evaluated with foreground masks applied (object-level).

| | UNISURF [20] / VolSDF [52] / NeuS [21] / Ours | | | |
| | PSNR ↑ | SSIM ↑ | LPIPS ↓ | Chamfer-$L_1$ ↓ |
|---|---|---|---|---|
| scan24 | 9.45 / 10.95 / 10.65 / **16.82** | 0.37 / 0.36 / 0.40 / **0.61** | 0.54 / 0.49 / 0.49 / **0.29** | 7.81 / 7.00 / 6.06 / **2.73** |
| scan37 | 7.19 / 11.17 / 10.07 / **13.44** | 0.22 / 0.35 / 0.30 / **0.45** | 0.33 / **0.21** / 0.27 / 0.22 | 7.54 / 6.95 / 7.24 / **4.92** |
| scan40 | 15.37 / 10.56 / 12.36 / **19.08** | 0.52 / 0.34 / 0.41 / **0.71** | 0.49 / 0.48 / 0.47 / **0.28** | 6.37 / 7.47 / 7.68 / **3.00** |
| scan55 | 12.71 / 16.35 / 15.93 / **19.55** | 0.30 / 0.58 / 0.43 / **0.73** | 0.39 / 0.21 / 0.28 / **0.17** | 8.38 / 2.90 / 5.85 / **2.37** |
| scan63 | 5.62 / 12.83 / 12.05 / **20.37** | 0.38 / 0.63 / 0.53 / **0.82** | 0.27 / 0.17 / 0.18 / **0.08** | 8.40 / 4.58 / 8.84 / **2.52** |
| scan65 | 11.23 / 17.76 / 18.10 / **19.00** | 0.57 / 0.81 / 0.79 / **0.86** | 0.18 / 0.08 / 0.09 / **0.07** | 5.08 / **2.30** / 4.65 / 2.71 |
| scan69 | 16.44 / 17.01 / 18.21 / **21.94** | 0.49 / 0.57 / 0.60 / **0.81** | 0.27 / 0.23 / 0.23 / **0.13** | 7.42 / 3.85 / 6.30 / **2.05** |
| scan83 | 7.62 / 12.72 / 13.52 / **22.63** | 0.37 / 0.49 / 0.52 / **0.85** | 0.19 / 0.15 / 0.13 / **0.05** | 7.92 / 9.14 / 9.62 / **3.14** |
| scan97 | 12.13 / 13.32 / 15.03 / **18.36** | 0.44 / 0.49 / 0.59 / **0.70** | 0.27 / 0.21 / 0.18 / **0.13** | 8.73 / 3.50 / 4.82 / **2.27** |
| scan105 | 12.12 / 14.60 / 15.94 / **21.48** | 0.46 / 0.54 / 0.62 / **0.80** | 0.29 / 0.25 / 0.21 / **0.15** | 8.89 / 6.52 / 8.19 / **3.61** |
| scan106 | 15.83 / 20.44 / 18.65 / **21.37** | 0.61 / 0.77 / 0.73 / **0.86** | 0.22 / 0.13 / 0.17 / **0.10** | 5.89 / **1.76** / 4.99 / 2.08 |
| scan110 | 14.89 / 13.69 / 17.68 / **19.79** | 0.51 / 0.39 / 0.70 / **0.79** | 0.19 / 0.26 / 0.14 / **0.11** | 7.68 / N/A / 5.75 / **2.03** |
| scan114 | 18.90 / 21.14 / 21.69 / **23.44** | 0.74 / 0.81 / 0.82 / **0.87** | 0.21 / 0.11 / 0.13 / **0.10** | 3.43 / **0.81** / 2.01 / 1.37 |
| scan118 | 16.09 / 21.03 / 18.29 / **27.44** | 0.48 / 0.74 / 0.61 / **0.91** | 0.24 / 0.14 / 0.16 / **0.05** | 6.47 / 3.93 / 6.16 / **1.83** |
| scan122 | 17.91 / 21.66 / 22.72 / **24.00** | 0.52 / 0.79 / 0.79 / **0.85** | 0.15 / 0.06 / 0.07 / **0.05** | 8.51 / **1.45** / 4.25 / 2.85 |
| mean | 12.90 / 15.68 / 16.06 / **20.58** | 0.47 / 0.58 / 0.59 / **0.77** | 0.28 / 0.21 / 0.21 / **0.13** | 7.23 / 4.44 / 6.16 / **2.63** |

