# OpenReview forum: "CorresNeRF: Image Correspondence Priors for Neural Radiance Fields"
_NeurIPS.cc/2023/Conference — NeurIPS 2023 poster_

### Official Review · Reviewer_1K2a · 2023-06-20

**Soundness:** 4 excellent
**Presentation:** 3 good
**Contribution:** 2 fair
**Rating:** 5
**Confidence:** 3

**Summary:**

The submission #8302, entitled "CorresNeRF: Image Correspondence Priors for Neural Radiance Fields" proposes a novel set of losses to improve the quality of NeRF under challenging conditions. In particular, the developed strategy effectively deals with the problem of sparse images. To achieve these performances, the authors take advantage of an out-of-the-box matching strategy between pairs of images to enforce geometric constraints during the training of the implicit representation. In particular, two types of losses are proposed to improve the quality of the reconstruction, namely the reprojection loss and the depth loss.
An extensive series of experiments demonstrates the relevance of these extra losses incorporated into the training. Another advantage of the proposed strategy is that it can easily be integrated into most implicit reconstruction techniques.

**Strengths:**

- The paper is well-written and straightforward
- The approach is simple and can easily be integrated into most NeRF-based approaches leading to improved results
- The proposed technique is very effective under sparse view constraints
- The computational overhead is very limited as matching strategies are often very fast

**Weaknesses:**

- The approach is very simple; the losses in themselves are not really new but demonstrate very effective results. The contributions appear to be limited, but the quality of the results might justify an acceptance. For this reason, I would like to express a mixed opinion regarding the acceptance of this work. Note that a relatively similar loss (on 3D structure obtained via correspondences) is applied in "Structure-Aware NeRF without Posed Camera via Epipolar Constraint" but with less success than in this manuscript #8302.
- CorressNeRF demonstrates good performance when few images are used, but it would be interesting to know the effect of these losses in more common scenarios.

**Questions:**

- An ablation study with an increasing number of images could be interesting to analyze how the method scales and if the demonstrated improvement are also valid with a larger density of images.
- Analysis with different matcher/keypoints would be a plus.
- What is the effect of the density of matched points? For instance, if a very sparse SIFT matching was used, what would be the expected effect of that?
- Exploring other types of loss would be interesting, for instance, some epipolar losses instead of the reprojection loss.

**Limitations:**

I have very little to say about this paper as it is very clear and straightforward. I would like to kindly recommend additional experiments, as explained in the previous section.

---

> ### Author Rebuttal · Authors · 2023-08-09
>
> ### Q1: Ablation study with an increasing number of images.
>
> We thank the reviewer for the suggestion. We tested the robustness of CorresNeRF with varying input view counts. Specifically, we doubled the number of input views from 3 to 6 and then evaluated CorresNeRF's performance on the LLFF dataset.
>
> | Method               | PSNR↑ | SSIM↑ | LPIPS↓ | Depth MAE↓ |
> | -------------------- | ----- | ----- | ------ | ---------- |
> | NeRF (3 views)       | 16.79 | 0.56  | 0.37   | 1.66       |
> | CorresNeRF (3 views) | 19.83 | 0.7   | 0.29   | 0.91       |
> | NeRF (6 views)       | 20.15 | 0.69  | 0.22   | 1.08       |
> | CorresNeRF (6 views) | 21.51 | 0.74  | 0.22   | 0.85       |
>
> The table indicates that CorresNeRF consistently outshines the baseline NeRF model, regardless of whether 3 or 6 views are used. Given that CorresNeRF is a plug-and-play module that can be added to any NeRF, provided quality image correspondences are available, its addition can boost the performance of NeRF, even in dense-view configurations.
>
> ### Q2: Analysis with different matcher/keypoints. Does it work with very sparse correspondence, e.g., SIFT matching?
>
> We thank the reviewer for the question. We assessed CorresNeRF's performance using correspondences derived from different image matching techniques, namely LoFTR [55] and DKMv3 [56]. The LLFF dataset served as our testing ground.
>
> | Method                  | PSNR↑ | SSIM↑ | LPIPS↓ | Depth MAE↓ |
> | ----------------------- | ----- | ----- | ------ | ---------- |
> | NeRF                    | 16.79 | 0.56  | 0.37   | 1.66       |
> | CorresNeRF (with LoFTR) | 18.13 | 0.64  | 0.30   | 1.10       |
> | CorresNeRF (with DKMv3) | 19.83 | 0.70  | 0.29   | 0.91       |
>
> The results show that CorresNeRF can outperform the vanilla NeRF, irrespective of the image matching technique employed, as long as quality correspondences are acquired. Additionally, CorresNeRF benefits from a "free" performance boost when a superior image matching method is employed, offering avenues for further enhancing CorresNeRF's performance.
>
> - [55] LoFTR: Detector-Free Local Feature Matching with Transformers, CVPR 2022
> - [56] DKM: Dense Kernelized Feature Matching for Geometry Estimation, CVPR 2023
>
> To study the effect of the density of matched points, we obtained correspondences using image matching methods and subsequently sampled a subset (50%, 25%, 12.5%, 6.25%, and 3.125%) of these correspondences to train CorresNeRF. We then assessed CorresNeRF's performance on the LLFF dataset.
>
> | Method                          | PSNR↑  | SSIM↑ | LPIPS↓ | Depth MAE↓ |
> | ------------------------------- | ------ | ----- | ------ | ---------- |
> | NeRF                            | 16.79  | 0.56  | 0.37   | 1.66       |
> | CorresNeRF (with 3.125% corres) | 18.616 | 0.647 | 0.322  | 1.129      |
> | CorresNeRF (with 6.25% corres)  | 18.934 | 0.657 | 0.299  | 1.113      |
> | CorresNeRF (with 12.5% corres)  | 18.854 | 0.66  | 0.287  | 1.108      |
> | CorresNeRF (with 25% corres)    | 19.068 | 0.669 | 0.269  | 1.10       |
> | CorresNeRF (with 50% corres)    | 18.986 | 0.67  | 0.266  | 1.085      |
> | CorresNeRF (with 100% corres)   | 19.83  | 0.70  | 0.29   | 0.91       |
>
> Notably, even with only 3.125% of the correspondences, CorresNeRF significantly outperforms the baseline NeRF model. The performance of CorresNeRF enhances as the correspondence quantity increases. When 100% of the correspondences are used, CorresNeRF achieves its peak performance. Thus, as long as quality correspondences are provided in adequate numbers, CorresNeRF can surpass the regular NeRF's performance.
>
> ### Q3: Exploring other types of loss, e.g. epipolar loss.
>
> We thank the reviewer for the suggestion. The epipolar loss, as discussed in the Structure-Aware NeRF paper, bears similarities to the reprojection loss used in CorresNeRF. We will delve deeper into the intricacies of the epipolar loss in the finalized version of our CorresNeRF paper.

---

> ### Author Response · Authors · 2023-08-14
>
> Dear Reviewer,
>
> We sincerely thank you for your precious time and efforts in reviewing our paper.
>
> We want to inquire whether our response has addressed your questions and concerns. We are more than happy to discuss with you further and provide additional materials.
>
> Best regards,
>
> Authors

---

> > ### Author Response · Authors · 2023-08-20
> >
> > Dear Reviewer,
> >
> > We sincerely thank you for your precious time and efforts in reviewing our paper.
> >
> > As we are approaching the deadline of the discussion period, we would like to inquire whether our response has addressed your questions and concerns. We are more than happy to discuss with you further and provide additional materials.
> >
> > Thank you again for the review and comments!
> >
> > Best regards,
> >
> > Authors

---

> > > ### Comment · Reviewer_1K2a · 2023-08-21
> > >
> > > First, I would like to sincerely thank the authors for their very detailed rebuttal, which addressed most of my concerns.
> > > Therefore, I would like to keep my positive opinion regarding the acceptance of this paper.

---

### Official Review · Reviewer_NREe · 2023-07-06

**Soundness:** 2 fair
**Presentation:** 3 good
**Contribution:** 2 fair
**Rating:** 5
**Confidence:** 5

**Summary:**

The paper presents NeRF regularization method for few view NeRF.

**Strengths:**

1. Using the state-of-the-art image matcher to regularize NeRF training is novel.
2. The paper is well-written and clear.

**Weaknesses:**

1. The paper proposes employing the cutting-edge image matcher to enhance NeRF training. However, a similar idea was presented in Neuris [1], which also suggested using patch matching to optimize NeRF training. Intuitively, one could assume that a state-of-the-art image matcher would identify more precise correspondences than patch match, leading to superior results. However, considering that Neuris integrates additional monocular depths and surface normals, the effectiveness of combining these three methods remains uncertain. Therefore, the author appears to have overlooked an essential baseline, Neuris. It is recommended that the author carry out experimental work based on the Neuris setting rather than implementing Neuris in their own setup, which make the conclusion more convincing.

2. The pixel loss and depth loss appear to aim towards the same goal. The concept of using both has been previously explored in DSAC [2], but was later discarded in DSAC++ [3], deemed as unnecessary. While the author provides an ablation study to illustrate the effectiveness of the reprojection loss, its value remains questionable. This is primarily because, in multiview settings, the reprojection loss mirrors the depth loss, making its unique contribution uncertain.

3. Minor, The citation of UNISURF in Figure 5 seems to be wrong.

[1] NeuRIS: Neural Reconstruction of Indoor Scenes Using Normal Priors
[2] DSAC-Differentiable RANSAC for Camera Localization
[3] Visual Camera Re-Localization from RGB and RGB-D Images Using DSAC

**Questions:**

Can the author show the proposed method is better than Neuris?

**Limitations:**

The image matcher sometimes fails when there are not enough overlap regions.

---

> ### Author Rebuttal · Authors · 2023-08-09
>
> ### Q1: Comparison with Neuris.
>
> We have examined the Neuris method and believe its approach of utilizing normal and monocular depth priors complements CorresNeRF. Notably, CorresNeRF serves as a plug-and-play module applicable to any NeRF, provided reasonable image correspondences can be achieved. Consequently, the image correspondence priors in CorresNeRF can integrate seamlessly with the normal and monocular depth priors in Neuris.
>
> Moreover, the Neuris paper predominantly focuses on indoor scenes, having been evaluated solely on the ScanNet dataset. It requires a monocular depth and surface normal models that are specifically trained for indoor settings. Conversely, CorresNeRF's image matching methods are versatile, catering to both indoor and outdoor scenes.
>
> ### Q2: Pixel loss and depth loss comparison.
>
> We thank the reviewer for the question. We believe that the pixel reprojection loss is more closely associated with the image matching method. This loss is defined in the 2D image space and is directly tied to the 2D image correspondences. In contrast, the depth loss is defined in the 3D space and has a strong connection to the NeRF model, given that the rendered depth in NeRF is the weighted sum of points along the camera ray. Based on our ablation study (Table 3 in the main paper), both the pixel loss and depth loss contribute significantly to the performance of CorresNeRF.
>
> ### Q3: Minor Citation issue of UNISURF.
>
> We thank the reviewer for pointing this out. We have fixed the citation issue.

---

> > ### Comment · Reviewer_NREe · 2023-08-15
> >
> > As I mentioned, Neuris also uses patch matching as prior. I think the author should show image matcher is better than patch match. If the method cannot outperform Neuris, the paper is meaningless. It is not convincing to say normal and monocular depth priors are compatible with CorresNeRF.
> >
> > I will maintain my rating unless I see concrete results.

---

> > > ### Author Response · Authors · 2023-08-18
> > > **Additional results and discussions on NeuRIS**
> > >
> > > ### Summary
> > >
> > > We would like to express our gratitude to the reviewer for the additional
> > > comments. To address these comments, we have conducted further experiments with
> > > NeuRIS using the DTU dataset under the same sparse view setting as described in
> > > the paper. In summary:
> > >
> > > - NeuRIS performs similarly to, or even worse than, the baseline NeuS method
> > >   upon which it is based. CorresNeRF significantly outperforms both NeuRIS and
> > >   NeuS.
> > > - The performance of NeuRIS is highly sensitive to the quality of the normal
> > >   priors. Since high-quality image correspondences are easier to obtain than
> > >   normal priors, we regard CorresNeRF as a more practical solution.
> > > - We carefully conduct the experiments to ensure a fair comparison between
> > >   CorresNeRF and NeuRIS. We have also visualized the normals for a more
> > >   intuitive comparison and analysis.
> > > - We consulted with the author of NeuRIS and received confirmation regarding our
> > >   observa- tions about the performance of NeuRIS on the DTU dataset.
> > >
> > > ### Quantitative Results
> > >
> > > We report the Chamfer-L1 distance results on the DTU dataset in the table below.
> > > All models were trained using the same three input views as described in the
> > > main paper, and the evaluation was done using the official DTU evaluation
> > > script. The values reported are Chamfer-L1 distances, where lower values are
> > > better.
> > >
> > > | Scene   | UNISURF | VolSDF   | NeuS | Neuris | Ours     |
> > > | ------- | ------- | -------- | ---- | ------ | -------- |
> > > | scan24  | 7.81    | 7.00     | 6.06 | 5.55   | **2.73** |
> > > | scan37  | 7.54    | 6.95     | 7.24 | 6.84   | **4.92** |
> > > | scan40  | 6.37    | 7.47     | 7.68 | 4.74   | **3.00** |
> > > | scan55  | 8.38    | 2.90     | 5.85 | 7.10   | **2.37** |
> > > | scan63  | 8.40    | 4.58     | 8.84 | N/A    | **2.52** |
> > > | scan65  | 5.08    | **2.30** | 4.65 | 5.48   | 2.71     |
> > > | scan69  | 7.42    | 3.85     | 6.30 | 7.81   | **2.05** |
> > > | scan83  | 7.92    | 9.14     | 9.62 | 11.28  | **3.14** |
> > > | scan97  | 8.73    | 3.50     | 4.82 | 5.82   | **2.27** |
> > > | scan105 | 8.89    | 6.52     | 8.19 | 7.60   | **3.61** |
> > > | scan106 | 5.89    | **1.76** | 4.99 | 7.72   | 2.08     |
> > > | scan110 | 7.68    | N/A      | 5.75 | 4.91   | **2.03** |
> > > | scan114 | 3.43    | **0.81** | 2.01 | 5.29   | 1.37     |
> > > | scan118 | 6.47    | 3.93     | 6.16 | 6.89   | **1.83** |
> > > | scan122 | 8.51    | **1.45** | 4.25 | 6.72   | 2.85     |
> > > | mean    | 7.23    | 4.44\*   | 6.16 | 6.70\* | **2.63** |
> > >
> > > \* Averaged over the valid results only.
> > >
> > > We observe that NeuRIS performs similarly to, or even worse than, the baseline
> > > NeuS method upon which it is based. This underperformance is likely due to the
> > > inaccuracy of the pre-trained normal priors (TiltedSN and SNU) when applied to
> > > the DTU dataset. We provide further visualizations and discussions in the
> > > following sections to elucidate this issue.
> > >
> > > ### Visualizations of the Normal Priors
> > >
> > > We propose that the performance of NeuRIS is highly sensitive to the quality of
> > > the normal priors. A key reason for NeuRIS’s poor performance on the DTU dataset
> > > seems to be the inaccuracy of these pre-trained normal priors.
> > >
> > > To further validate this hypothesis, we visualize the normal priors computed by
> > > TiltedSN on the DTU dataset. As shown in Figure 1, while most of the normal
> > > predictions are reasonable, a significant number of normals point in tilted or
> > > incorrect directions. This discrepancy is likely due to TiltedSN being
> > > pre-trained on indoor scene datasets, which differ substantially from the
> > > object-level DTU dataset.
> > >
> > > We also provide visualizations of the normal priors on the ScanNet dataset,
> > > which are used in NeuRIS. These visualizations are shown in Figure 2.
> > >
> > > ### Replies from the Author of NeuRIS
> > >
> > > To further investigate the performance of NeuRIS on the DTU dataset, we
> > > contacted the first author of NeuRIS. The author provided us with the following
> > > reply:
> > >
> > > > "The two pre-trained normal priors (TiltedSN and SNU) are indeed not very
> > > > accurate on the DTU dataset. Whether the performance will be better or worse
> > > > (compared to the baseline Neus) may depend on other factors, such as the
> > > > number of sparse views used in the experiment."
> > >
> > > The NeuRIS author also reviewed our visualizations of the DTU normal priors and
> > > confirmed the correctness of our implementation and our observations.
> > >
> > > Furthermore, we found that the NeuRIS codebase contains partial code
> > > for loading DTU data. However, the author neither reported the evaluation
> > > results for the DTU dataset in the paper, nor provided the full configurations
> > > and documentation necessary to run experiments using the DTU dataset in their
> > > codebase.

---

> > > > ### Comment · Reviewer_NREe · 2023-08-18
> > > >
> > > > Thanks for the author's comments and I increase my score, but the author violated Neurips policy this year. The external link is prohibited. Please see the emailed instructions. The link should be sent to AC first and only for videos.

---

> > > > > ### Author Response · Authors · 2023-08-19
> > > > >
> > > > > Dear reviewer,
> > > > >
> > > > > We thank the reviewer for the additional comments. As suggested, we have removed the anonymous link according to the new policy.
> > > > >
> > > > > We thank the reviewer again for considering our additional experiments on Neuris and agreeing to increase the paper's score. We kindly remind the reviewer to update the scores accordingly in the system.
> > > > >
> > > > > Best regards,
> > > > > Authors

---

> > > > > > ### Author Response · Authors · 2023-08-20
> > > > > >
> > > > > > Dear reviewer,
> > > > > >
> > > > > > We sincerely thank you for reviewing and considering our additional materials regarding Neuris, and updating the score according to the new materials.
> > > > > >
> > > > > > It's been our pleasure to conduct these new experiments, and we believe they will contribute to the final value of the paper. Thanks again for the review and the suggestions.
> > > > > >
> > > > > > Best regards,
> > > > > >
> > > > > > Authors

---

> ### Author Response · Authors · 2023-08-14
>
> Dear Reviewer,
>
> We sincerely thank you for your precious time and efforts in reviewing our paper.
>
> We want to inquire whether our response has addressed your questions and concerns. We are more than happy to discuss with you further and provide additional materials.
>
> Best regards,
>
> Authors

---

### Official Review · Reviewer_7TAK · 2023-07-06

**Soundness:** 2 fair
**Presentation:** 2 fair
**Contribution:** 2 fair
**Rating:** 4
**Confidence:** 3

**Summary:**

This paper proposed CorresNeRF, a method that leverages image correspondence priors to improve NeRF training on sparse input views. The correspodence matching is computed by off-the-shelf methods. The authors augue that the introduced inexpensive image correspodence priors can be used to supervise training of arbitrary NeRFs and lead to better performance / faster convergence taking sparse view inputs. Further, a robust correspondence loss is designed, including reprojection loss and depth loss baesd on correspondence priors. Overall, the method demonstrates superior reconstruction quality against baselines like VolSDF and NeuS, etc.

**Strengths:**

The paper's main contributions are summarized as follows:

- Introduction of image correspondence as a cost-effective prior to supervise the training of any NeRFs.
- Design of a pipeline to obtain robust correspondences from standard methods, including automatic augmentation and filtering.
- Introduce a robust correspondence loss incorporating reprojection loss and depth loss based on correspondence priors.
- Extensive experiments conducted on various baselines and datasets demonstrating the method's effectiveness across different types of neural implicit representations.

The authors conduct extensive experiments on various datasets, which demonstrate the effectiveness of their method. They find significant improvements in both novel view synthesis and surface reconstruction metrics. The proposed method outperforms other state-of-the-art sparse-view reconstruction methods and works well with various types of NeRF, including those with other priors.

**Weaknesses:**

There are some limitations that should be properly discussed:

- Dependence on the quality of image correspodence. The quality of the obtained image correspodence matching significantly impacts the effectiveness of the proposed approach. Less accurate correspondences can negatively affect the supervised training of NeRF, which might lead to suboptimal reconstruction results.
- Performance in non-sparse scenarios. The method is focused on the advantage of using CorresNeRF in sparse-view configurations, but it doesn't mention how this method would perform in dense-view configurations. It would be interesting to show such an ablation study to verify this.
- The main comparisons show VolSDF and NeuS results as baselines. However, a more adequate baseline would be SparseNeuS:
SparseNeuS: Fast Generalizable Neural Surface Reconstruction from Sparse Views. ECCV 2022.
- The method shows 3 input views on DTU/LLFF datasets. What happen if arbitrary number of input views are given? This is soemwhat related to the 2nd point above. But it would be nice to have such experiments to better evaluate the robustness of the proposed pipeline.


**Questions:**

Some questions:

- Clarification on the image correspondence methods.The paper mentioned that image correspondences are obtained via off-the-shelf method. Could you provide more details on how they are acquired and how this (i.e. selection of the method) would affect the final reconstruction quality?

- Robustness to the noise/outliers in the correspondence. An automatic augmentation and outlier removal process was designed, which seems to be key to the proposed approach. It would be useful to provide more details on the robustness of the design with some various level of noise/outliers on some scenes.

-  What is the minimum quality of correspondence necessary for the method to outperform traditional NeRF implementations?

- Impact of loss terms on results. How did the inclusion of correspondence pixel reprojection loss and correspondence depth loss affect the final results? Could you clarify how these losses contribute to the reconstruction quality?

- Extreme and failure cases. One such case I'd imagine is glossy/specular or textureless surfaces.


**Limitations:**

Limitations are discussed in the above weakness section.

---

> ### Author Rebuttal · Authors · 2023-08-09
>
>
> ### Q1 (from weaknesses): Dependence on the quality of image correspondence.
>
> We thank the reviewer for the suggestion. We employed image matching methods to obtain correspondences and subsequently introduced Gaussian noise to these correspondences. Specifically, we added Gaussian noise with standard deviations of 1, 2, and 4 pixels to both x and y pixel coordinates of the correspondences. We then assessed the performance of CorresNeRF using the LLFF dataset.
>
> Section 3.2 describes how CorresNeRF employs an automatic outlier removal process based on camera reprojection error. Column 2 in the table below reports the relative number of correspondences remaining after this filtering process.
>
> | Method                       | Corres # After Auto Filter | PSNR↑ | SSIM↑ | LPIPS↓ | Depth MAE↓ |
> | ---------------------------- | -------------------------- | ----- | ----- | ------ | ---------- |
> | NeRF                         | 0.00%                      | 16.79 | 0.56  | 0.37   | 1.66       |
> | CorresNeRF (noise_std = 4px) | 13.96%                     | 18.31 | 0.61  | 0.48   | 1.04       |
> | CorresNeRF (noise_std = 2px) | 27.04%                     | 19.16 | 0.66  | 0.33   | 1.06       |
> | CorresNeRF (noise_std = 1px) | 48.91%                     | 19.31 | 0.67  | 0.28   | 1.06       |
> | CorresNeRF (noise_std = 0px) | 100.00%                    | 19.83 | 0.70  | 0.29   | 0.91       |
>
> When image correspondences are contaminated by noise, the automatic outlier removal process discards more of them, resulting in fewer correspondences for CorresNeRF to utilize. However, the correspondences that remain are deemed higher quality. Consequently, CorresNeRF maintains satisfactory performance even with noisy correspondences.
>
> ### Q2 (from weaknesses): Performance in non-sparse scenarios.
>
> We tested the robustness of CorresNeRF with varying input view counts. Specifically, we doubled the number of input views from 3 to 6 and then evaluated CorresNeRF's performance on the LLFF dataset.
>
> | Method               | PSNR↑ | SSIM↑ | LPIPS↓ | Depth MAE↓ |
> | -------------------- | ----- | ----- | ------ | ---------- |
> | NeRF (3 views)       | 16.79 | 0.56  | 0.37   | 1.66       |
> | CorresNeRF (3 views) | 19.83 | 0.7   | 0.29   | 0.91       |
> | NeRF (6 views)       | 20.15 | 0.69  | 0.22   | 1.08       |
> | CorresNeRF (6 views) | 21.51 | 0.74  | 0.22   | 0.85       |
>
> The table indicates that CorresNeRF consistently outshines the baseline NeRF model, regardless of whether 3 or 6 views are used. Given that CorresNeRF is a plug-and-play module that can be added to any NeRF, provided quality image correspondences are available, its addition can boost the performance of NeRF, even in dense-view configurations.
>
> ### Q3 (from weaknesses): Comparison with other baselines, such as SparseNeuS.
>
> In summary, as long as reliable image correspondences can be established, CorresNeRF can serve as a generic plug-and-play module to enhance the performance of any NeRF model, including SparseNeuS.
>
> SparseNeuS learns generalizable priors from image features, encoding coarse-to-fine geometry volumes for generic surface prediction. These methods are distinct from the image correspondence priors used in CorresNeRF. Consequently, SparseNeuS can be integrated with CorresNeRF to further enhance performance. We plan to incorporate additional experiments combining CorresNeRF with SparseNeuS in the final version of the paper.
>
> ### Q4 (from questions): Clarification on the image correspondence methods, including the selection of the method.
>
> We assessed CorresNeRF's performance using correspondences derived from different image matching techniques, namely LoFTR [55] and DKMv3 [56]. The LLFF dataset served as our testing ground.
>
> | Method                  | PSNR↑ | SSIM↑ | LPIPS↓ | Depth MAE↓ |
> | ----------------------- | ----- | ----- | ------ | ---------- |
> | NeRF                    | 16.79 | 0.56  | 0.37   | 1.66       |
> | CorresNeRF (with LoFTR) | 18.13 | 0.64  | 0.30   | 1.10       |
> | CorresNeRF (with DKMv3) | 19.83 | 0.70  | 0.29   | 0.91       |
>
> The results show that CorresNeRF can outperform the vanilla NeRF, irrespective of the image matching technique employed, as long as quality correspondences are acquired. Additionally, CorresNeRF benefits from a "free" performance boost when a superior image matching method is employed, offering avenues for further enhancing CorresNeRF's performance.
>
> ### Q5 (from questions): Robustness to the noise/outliers in the correspondence, and what is the minimum quality of correspondence necessary for the method to outperform traditional NeRF implementations?
>
> Please kindly refer the answer above for Q1.
>
> ### Q6 (from questions): Impact of loss terms. How do pixel reprojection loss and correspondence depth loss affect the final results?
>
> We believe that the pixel reprojection loss is more closely associated with the image matching method. This loss is defined in the 2D image space and is directly tied to the 2D image correspondences. In contrast, the depth loss is defined in the 3D space and has a strong connection to the NeRF model, given that the rendered depth in NeRF is the weighted sum of points along the camera ray. Based on our ablation study (Table 3 in the main paper), both the pixel loss and depth loss contribute significantly to the performance of CorresNeRF.
>
> ### Q7 (from questions): Extreme and failure cases
>
> As described in the limitations section of the paper, CorresNeRF is dependent on the results produced by the image matching method. If the input surface is glossy, specular, or lacks texture, the image matching method may fail to generate accurate correspondences. Moreover, such correspondences could be eliminated by the automated outlier removal process, which is based on camera reprojection errors. In these scenarios, the enhancements offered by CorresNeRF over the baseline NeRF model might be minimal.

---

> > ### Author Response · Authors · 2023-08-20
> >
> > Dear Reviewer,
> >
> > We sincerely thank you for your precious time and efforts in reviewing our paper.
> >
> > As we are approaching the deadline of the discussion period, we would like to inquire whether our response has addressed your questions and concerns. We are more than happy to discuss with you further and provide additional materials.
> >
> > Thank you again for the review and comments!
> >
> > Best regards,
> >
> > Authors

---

> ### Author Response · Authors · 2023-08-14
>
> Dear Reviewer,
>
> We sincerely thank you for your precious time and efforts in reviewing our paper.
>
> We want to inquire whether our response has addressed your questions and concerns. We are more than happy to discuss with you further and provide additional materials.
>
> Best regards,
>
> Authors

---

### Official Review · Reviewer_cKcF · 2023-07-07

**Soundness:** 2 fair
**Presentation:** 2 fair
**Contribution:** 2 fair
**Rating:** 4
**Confidence:** 3

**Summary:**

The paper introduces CorresNeRF, a method that leverages image correspondence priors to improve the performance of Neural Radiance Fields (NeRF) in scenarios with sparse input views. The authors propose a plug-and-play module that incorporates correspondence priors into the training process by adding loss terms on the reprojection error and depth error of the correspondence points. They develop an adaptive algorithm for augmenting and filtering the correspondence priors to enhance their quality. The proposed method is evaluated on novel view synthesis and surface reconstruction tasks using density-based and SDF-based neural implicit representations across different datasets.

The proposed CorresNeRF utilizes image correspondence priors to supervise the training of NeRF models. This approach addresses the challenge of sparse input views and enhances the performance of NeRF in reconstructing 3D geometries.
The authors propose an automatic augmentation and outlier removal process for improving the quality and robustness of the correspondence priors. This process enhances the dense correspondence estimation and mitigates the effects of inaccurate correspondences.
The paper formulates a correspondence loss that incorporates reprojection and depth errors based on the correspondence priors. This loss effectively guides the learning of implicit functions in NeRF models and improves their performance.

**Strengths:**

The paper demonstrates several strengths across different dimensions:

The paper introduces the concept of leveraging image correspondence priors to improve the performance of NeRF models in sparse-view scenarios. This novel approach addresses the challenge of reconstructing 3D geometries with limited input views and introduces the use of image correspondences as explicit supervision for learning implicit functions in NeRF. The combination of image correspondence priors and NeRF training is a creative and innovative approach that expands the capabilities of NeRF models.

The paper addresses a significant problem in the field of 3D reconstruction and view synthesis. Sparse-view scenarios are common in real-world applications, and improving the performance of NeRF models under such conditions has practical implications. The proposed method offers a practical and effective solution by leveraging image correspondence priors, which are readily obtainable and can be computed using standard methods. The experimental results demonstrate the superiority of the proposed approach over previous methods, highlighting its potential for advancing the state-of-the-art in novel view synthesis and surface reconstruction tasks.

The paper presents a well-designed methodology with clear objectives and a systematic evaluation process. The authors carefully consider the limitations of existing methods and propose solutions to overcome them. The proposed CorresNeRF method incorporates robust correspondence loss and automatic augmentation and filtering of correspondence priors, enhancing the quality and effectiveness of the training process. The experimental evaluation is thorough, encompassing various neural implicit representations and datasets, and the results demonstrate significant improvements in performance metrics.

The proposed method is described in a structured manner, with detailed explanations of the augmentation and filtering process, formulation of correspondence loss, and evaluation metrics. The figures and equations further enhance the clarity of the paper, aiding in the understanding of the concepts and techniques presented.


**Weaknesses:**

While the paper demonstrates several strengths, there are also a few areas where it could be improved:

Experimental Evaluation: The paper would benefit from a more detailed analysis of the computational efficiency and resource requirements of the CorresNeRF method. Providing insights into the computational demands and resource utilization of the approach would help readers understand the practical implications and scalability of the method.

Image Correspondences: While the paper introduces image correspondences as priors, it is important to acknowledge the potential challenges in estimating accurate and robust image correspondences, especially in scenarios with occluded or noisy images. Conducting a sensitivity analysis of correspondence accuracy would provide a clearer understanding of the method's performance under different conditions and shed light on its robustness and generalization capabilities.

Comparison with State-of-the-Art: The paper would benefit from a more comprehensive comparison with existing state-of-the-art methods for sparse-view reconstruction, such as MVSNeRF and GeoNeRF. Providing a thorough evaluation and comparison against these methods would help establish the superiority and novelty of the proposed CorresNeRF method.

**Questions:**

Could you provide a more detailed analysis of the computational efficiency and resource requirements of the CorresNeRF method? Specifically, it would be valuable to include information on training time, inference speed, and memory utilization to understand the practical implications and scalability of the approach.

Estimating accurate and robust image correspondences can be challenging in real-world scenarios. It would be helpful to discuss the performance under inaccurate correspondence.

It would be beneficial to provide a more comprehensive comparison with existing state-of-the-art methods for sparse-view reconstruction, such as MVSNeRF and GeoNeRF. Including a thorough evaluation and comparison against these methods would strengthen the justification for the superiority and novelty of the proposed CorresNeRF method.

**Limitations:**

The limitation is discussed.

---

> ### Author Rebuttal · Authors · 2023-08-09
>
> ### Q1: Experimental Evaluation: computational efficiency of CorresNeRF.
>
> We thank the reviewer for the question. At inference time, CorresNeRF operates at exactly the same runtime as the baseline NeRF model. However, during training, CorresNeRF incurs additional runtime overheads due to the search for correspondences and the computation of the correspondence loss.
>
> We've conducted a supplementary runtime analysis experiment. Specifically, we evaluated the runtime for both the forward pass (rendering) and backward pass (gradient computation), performing these measurements over 100 iterations with a batch size of 1024 on the fern scene of the LLFF dataset. Testing was conducted on a single NVIDIA RTX 2080Ti GPU, and both average and standard deviation of the runtimes were reported.
>
> | Method     | Training Forward (ms) | Training Backward (ms) |
> | ---------- | --------------------- | ---------------------- |
> | NeRF       | 51.722 ± 0.312        | 70.941 ± 0.429         |
> | CorresNeRF | 126.896 ± 3.254       | 124.160 ± 2.371        |
>
> It's worth noting that the runtime overhead of CorresNeRF is contingent upon the ratio of pixels possessing valid correspondences. We believe the additional overhead introduced by CorresNeRF is justifiable, especially given the substantial performance enhancement over the baseline NeRF model.
>
> ### Q2: Image Correspondences: sensitivity analysis of image correspondences
>
> We thank the reviewer for the suggestion. We conducted two sets of supplementary experiments to assess the impact of 1) quality and 2) quantity of image correspondences on CorresNeRF's performance. Our results indicate that CorresNeRF maintains impressive performance even when faced with noisy correspondences or when utilizing only a small subset of image correspondences. This underscores the robustness of CorresNeRF; it demonstrates that as long as reasonable correspondences are present, CorresNeRF can amplify the performance over the baseline NeRF model.
>
> **Effects of correspondence quality (robustness to noise)**
>
> We employed image matching methods to obtain correspondences and subsequently introduced Gaussian noise to these correspondences. Specifically, we added Gaussian noise with standard deviations of 1, 2, and 4 pixels to both x and y pixel coordinates of the correspondences. We then assessed the performance of CorresNeRF using the LLFF dataset.
>
> Section 3.2 describes how CorresNeRF employs an automatic outlier removal process based on camera reprojection error. Column 2 in the table below reports the relative number of correspondences remaining after this filtering process.
>
> | Method                       | Corres # After Auto Filter | PSNR↑ | SSIM↑ | LPIPS↓ | Depth MAE↓ |
> | ---------------------------- | -------------------------- | ----- | ----- | ------ | ---------- |
> | NeRF                         | 0.00%                      | 16.79 | 0.56  | 0.37   | 1.66       |
> | CorresNeRF (noise_std = 4px) | 13.96%                     | 18.31 | 0.61  | 0.48   | 1.04       |
> | CorresNeRF (noise_std = 2px) | 27.04%                     | 19.16 | 0.66  | 0.33   | 1.06       |
> | CorresNeRF (noise_std = 1px) | 48.91%                     | 19.31 | 0.67  | 0.28   | 1.06       |
> | CorresNeRF (noise_std = 0px) | 100.00%                    | 19.83 | 0.70  | 0.29   | 0.91       |
>
> When image correspondences are contaminated by noise, the automatic outlier removal process discards more of them, resulting in fewer correspondences for CorresNeRF to utilize. However, the correspondences that remain are deemed higher quality. Consequently, CorresNeRF maintains satisfactory performance even with noisy correspondences.
>
> **Effects of correspondence quantity**
>
> We obtained correspondences using image matching methods and subsequently sampled a subset (50%, 25%, 12.5%, 6.25%, and 3.125%) of these correspondences to train CorresNeRF. We then assessed CorresNeRF's performance on the LLFF dataset.
>
> | Method                          | PSNR↑  | SSIM↑ | LPIPS↓ | Depth MAE↓ |
> | ------------------------------- | ------ | ----- | ------ | ---------- |
> | NeRF                            | 16.79  | 0.56  | 0.37   | 1.66       |
> | CorresNeRF (with 3.125% corres) | 18.616 | 0.647 | 0.322  | 1.129      |
> | CorresNeRF (with 6.25% corres)  | 18.934 | 0.657 | 0.299  | 1.113      |
> | CorresNeRF (with 12.5% corres)  | 18.854 | 0.66  | 0.287  | 1.108      |
> | CorresNeRF (with 25% corres)    | 19.068 | 0.669 | 0.269  | 1.10       |
> | CorresNeRF (with 50% corres)    | 18.986 | 0.67  | 0.266  | 1.085      |
> | CorresNeRF (with 100% corres)   | 19.83  | 0.70  | 0.29   | 0.91       |
>
> Notably, even with only 3.125% of the correspondences, CorresNeRF significantly outperforms the baseline NeRF model. The performance of CorresNeRF enhances as the correspondence quantity increases. When 100% of the correspondences are used, CorresNeRF achieves its peak performance. Thus, as long as quality correspondences are provided in adequate numbers, CorresNeRF can surpass the regular NeRF's performance.
>
> ### Q3: Comparison with the state-of-the-art (MVSNeRF and GeoNeRF)
>
> We thank the reviewer for the suggestion. In essence, if reasonable image correspondences can be secured, CorresNeRF can serve as a versatile plug-and-play module to enhance any NeRF model, including MVSNeRF and GeoNeRF.
>
> MVSNeRF and GeoNeRF are expansive NeRF models suitable for sparse-view settings. MVSNeRF capitalizes on a plane-sweeping cost volume, while GeoNeRF constructs cost volumes through transformer-based feature aggregation. These methodologies are distinct from the image correspondence priors employed by CorresNeRF. Hence, combining these techniques with CorresNeRF could potentially yield further performance improvements. We plan to incorporate additional experiments, merging CorresNeRF with MVSNeRF and GeoNeRF, in the finalized version of the paper.

---

> ### Author Response · Authors · 2023-08-14
>
> Dear Reviewer,
>
> We sincerely thank you for your precious time and efforts in reviewing our paper.
>
> We want to inquire whether our response has addressed your questions and concerns. We are more than happy to discuss with you further and provide additional materials.
>
> Best regards,
>
> Authors

---

> > ### Author Response · Authors · 2023-08-20
> >
> > Dear Reviewer,
> >
> > We sincerely thank you for your precious time and efforts in reviewing our paper.
> >
> > As we are approaching the deadline of the discussion period, we would like to inquire whether our response has addressed your questions and concerns. We are more than happy to discuss with you further and provide additional materials.
> >
> > Thank you again for the review and comments!
> >
> > Best regards,
> >
> > Authors

---

> ### Comment · Reviewer_cKcF · 2023-08-22
>
> The training speed of NeRF will be significantly slower when incorporating the proposed correspondence design, which will be a much bigger problem for the current fashion of fast scene representation (e.g., Instant-NGP).
> With the high correspondence computation, I believe the fast speed merit of Instant-NGP or other fast models will be lost completely.
> The application scenario and practical value of this method is relatively narrow. I keep my rating of borderline reject.

---

> > ### Author Response · Authors · 2023-08-22
> >
> > We appreciate the reviewer's insights and the introduction of Instant-NGP and other fast models into our discussion.
> >
> > In response to the concerns raised:
> >
> > > Reviewer comment: "With the high correspondence computation, I believe the fast speed merit of Instant-NGP or other fast models will be lost completely."
> >
> > The above statement is not true. **In fact, the speed advantages of Instant-NGP are still retained.** For instance, if CorresNeRF takes 2.x times the training time of the standard NeRF, the "Corres-Instant-NGP" will similarly take 2.x times the training time of Instant-NGP, preserving the speed advantage of Instant-NGP.
> >
> > In CorresNeRF, image correspondences are pre-computed and cached. The additional runtime mainly comes from the extra forward/backward pass for corresponding pixels. As such, the runtime overhead shall be calculated on a **"relative scale"** rather than an "absolute scale".
> >
> > Furthermore, CorresNeRF introduces **zero inference overhead** but offers a superior reconstruction quality.
> >
> > > Reviewer comment: "The application scenario and practical value of this method is relatively narrow."
> >
> > It's important to highlight that CorresNeRF and Instant-NGP address **orthogonal challenges**. While Instant-NGP emphasizes rapid training, the primary goal of CorresNeRF is to enhance reconstruction quality in sparse-view contexts. Given the orthogonal design considerations of these methods, a direct comparison of their training times may not be meaningful.
> >
> > Moreover, we anticipate that Instant-NGP's performance would be much worse than that of CorresNeRF in sparse-view conditions, such as in 3-view or 6-view scenarios.

---

### Official Review · Reviewer_yxQE · 2023-07-20

**Soundness:** 3 good
**Presentation:** 3 good
**Contribution:** 3 good
**Rating:** 5
**Confidence:** 4

**Summary:**

The paper proposes an approach for sparse-view NeRF reconstruction by using image correspondences as a prior. NeRF under the sparse-view regime is overparameterized and under constrained hence requiring a prior to optimize. This paper proposes to use image correspondences that are extracted across the different views, in particular, they use DKMv3. They propose two additional loss functions based on the correspondences from the prior: the first uses the reprojection error by using the expected depth as predicted by the NeRF while the second uses a correspondence-depth based loss by finding the closest 3D points in space based on the correspondence from the prior. Experiments on novel view synthesis and surface reconstruction show the improvement of their proposed approach.

**Strengths:**

The paper proposes to use image correspondences as a prior for sparse view nerf reconstruction which is intuitive and sound. Image correspondences as a prior is generalizable and hence a pretrained model can be used, namely DKMv3. They propose two simple, yet intuitive losses for their approach. Experiments show that the proposed method is able to perform better compared to existing baselines.

**Weaknesses:**

The effectiveness of the method is relying on the accurate prediction of the correspondences, and it is known that correspondences can be erroneous on texture less regions, illumination changes or wide-baseline cameras. On real scenes, these issues might arise more -- e.g. on sparse scannet images as used by existing benchmarks [53, 54]. The sparse view inputs here have wide-camera baselines as opposed to forward facing scenes in LLFF. It would be more convincing if the method can also perform reasonably in such settings.


Some references on sparse view NeRF:

**[53]** Dense Depth Priors for Neural Radiance Fields from Sparse Input Views, CVPR '22
**[54]** SCADE: NeRFs from Space Carving with Ambiguity-Aware Depth Estimates, CVPR '23

**Questions:**

1. What happens when other image correspondence priors (not DKMv3) is used? It will be beneficial to see how this would affect results, for example for both another neural network based prior as well as a handcrafted prior, e.g. output of COLMAP, even if it will only give correspondences at sparse pixel locations.
2. NeRF depth is used for the correspondence pixel reprojection loss, and it is known that in the sparse regime the NeRF depth can be erroneous. Did this cause an issue in the convergence of the network? Would it have helped to add in a depth loss supervision as a prior?

**Limitations:**

The authors have included limitations in the main paper of the submission.

---

> ### Author Rebuttal · Authors · 2023-08-09
>
> ### Q1: Performance on wide-camera baselines (e.g. ScanNet) in addition to forward-facing LLFF.
>
> We appreciate the reviewer's inquiry. In our paper, we present evaluations of
> CorresNeRF on the LLFF dataset with forward-facing cameras, as well as on the
> DTU dataset where the cameras have a spherical configuration.
>
> In "wide-camera" scenarios such as the ScanNet dataset, establishing
> correspondences between input images necessitates a significant overlap between
> images containing textured regions. Provided that reasonable correspondences can
> be achieved through image matching techniques, CorresNeRF remains effective. We
> intend to include additional experiments on the ScanNet dataset in the paper's
> final version.
>
> In essence, CorresNeRF depends on image matchers to ascertain correspondences.
> As long as these correspondences are established, CorresNeRF can function as a
> generic plug-and-play module, enhancing the efficacy of any NeRF model, which
> bodes well for the broader community.
>
> ### Q2: What happens when other image correspondence priors is used?
>
> We thank the reviewer for the question. Indeed, the quality and quantity of
> image correspondences influence CorresNeRF's performance. We've introduced a new
> experiment where we compare CorresNeRF's performance when using different image
> matching techniques, specifically LoFTR [55] and DKMv3 [56], on the LLFF
> dataset. Here are the results:
>
> | Method                  | PSNR↑ | SSIM↑ | LPIPS↓ | Depth MAE↓ |
> | ----------------------- | ----- | ----- | ------ | ---------- |
> | NeRF                    | 16.79 | 0.56  | 0.37   | 1.66       |
> | CorresNeRF (with LoFTR) | 18.13 | 0.64  | 0.30   | 1.10       |
> | CorresNeRF (with DKMv3) | 19.83 | 0.70  | 0.29   | 0.91       |
>
> It's evident that regardless of the chosen image matching method, if viable
> correspondences are established, CorresNeRF can enhance performance beyond the
> original NeRF. Additionally, when a superior image matching method is employed,
> CorresNeRF naturally benefits, potentially driving its performance further.
>
> Regarding COLMAP correspondences, we applied COLMAP to the LLFF dataset using
> specified cameras in 3 sparse views. Here's a summary:
>
> | Scene    | COLMAP: Num of Corres (Pixel Coverage %) | DKMv3: Num of Corres (Pixel Coverage %) |
> | -------- | ---------------------------------------- | --------------------------------------- |
> | fern     | 362 (0.19%)                              | 368,798 (57%)                           |
> | flower   | 685 (0.35%)                              | 356,044 (73%)                           |
> | fortress | 609 (0.31%)                              | 430,044 (76%)                           |
> | horns    | 512 (0.27%)                              | 271,705 (47%)                           |
> | leaves   | 201 (0.11%)                              | 198,412 (48%)                           |
> | orchids  | 229 (0.12%)                              | 242,620 (37%)                           |
> | room     | 345 (0.18%)                              | 260,308 (40%)                           |
> | trex     | 644 (0.34%)                              | 233,950 (35%)                           |
>
> From this data, it's clear that COLMAP yields sparser correspondences compared
> to DKMv3. While CorresNeRF's performance might be restricted with only a few
> correspondences, it still outperforms the standard NeRF model. We direct the
> reviewer to Table 1 in the supplementary material for more detailed statistics.
>
> - [55] LoFTR: Detector-Free Local Feature Matching with Transformers, CVPR 2022
> - [56] DKM: Dense Kernelized Feature Matching for Geometry Estimation, CVPR 2023
>
> ### Q3: NeRF depth can be erroneous in the sparse regime. Did this cause an issue in the convergence of the network? Would it have helped to add in a depth loss supervision as a prior?
>
> We thank the reviewer for the question. Indeed, NeRF's depth can be unreliable
> in a sparse regime without an auxiliary prior. In CorresNeRF, image
> correspondences are used as priors, ensuring the depth values are tethered by
> the corres depth loss and pixel reprojection loss. This makes the depth values
> more reliable than those in the standard NeRF model.
>
> Regarding the proposition of introducing depth loss supervision as a prior, we
> believe that CorresNeRF's corres depth loss and pixel reprojection loss already
> implicitly offer such supervision. Given the camera parameters and
> correspondences, depth values can be triangulated, and CorresNeRF's loss terms
> appropriately model this relationship.

---

> > ### Author Response · Authors · 2023-08-14
> >
> > Dear Reviewer,
> >
> > We sincerely thank you for your precious time and efforts in reviewing our paper.
> >
> > We want to inquire whether our response has addressed your questions and concerns. We are more than happy to discuss with you further and provide additional materials.
> >
> > Best regards,
> >
> > Authors

---

> > > ### Author Response · Authors · 2023-08-20
> > >
> > > Dear Reviewer,
> > >
> > > We sincerely thank you for your precious time and efforts in reviewing our paper.
> > >
> > > As we are approaching the deadline of the discussion period, we would like to inquire whether our response has addressed your questions and concerns. We are more than happy to discuss with you further and provide additional materials.
> > >
> > > Thank you again for the review and comments!
> > >
> > > Best regards,
> > >
> > > Authors

---

### Author Rebuttal · Authors · 2023-08-10

We thank all the reviewers for their insightful comments and questions. In this section, we provide a summary of our responses and present new experimental results. Specifically, we introduce new experiments that examine:

- The robustness of CorresNeRF with noisy correspondences
- The robustness of CorresNeRF when using fewer correspondences
- The performance of CorresNeRF with additional input views (3-view and 6-view)
- The performance of CorresNeRF when using different image matchers

## Robustness of CorresNeRF with Noisy Correspondences

We employed image matching methods to obtain correspondences and subsequently introduced Gaussian noise to these correspondences. Specifically, we added Gaussian noise with standard deviations of 1, 2, and 4 pixels to both x and y pixel coordinates of the correspondences. We then assessed the performance of CorresNeRF using the LLFF dataset.

Section 3.2 describes how CorresNeRF employs an automatic outlier removal process based on camera reprojection error. Column 2 in the table below reports the relative number of correspondences remaining after this filtering process.

| Method                       | Corres # After Auto Filter | PSNR↑ | SSIM↑ | LPIPS↓ | Depth MAE↓ |
| ---------------------------- | -------------------------- | ----- | ----- | ------ | ---------- |
| NeRF                         | 0.00%                      | 16.79 | 0.56  | 0.37   | 1.66       |
| CorresNeRF (noise_std = 4px) | 13.96%                     | 18.31 | 0.61  | 0.48   | 1.04       |
| CorresNeRF (noise_std = 2px) | 27.04%                     | 19.16 | 0.66  | 0.33   | 1.06       |
| CorresNeRF (noise_std = 1px) | 48.91%                     | 19.31 | 0.67  | 0.28   | 1.06       |
| CorresNeRF (noise_std = 0px) | 100.00%                    | 19.83 | 0.70  | 0.29   | 0.91       |

When image correspondences are contaminated by noise, the automatic outlier removal process discards more of them, resulting in fewer correspondences for CorresNeRF to utilize. However, the correspondences that remain are deemed higher quality. Consequently, CorresNeRF maintains satisfactory performance even with noisy correspondences.

## Robustness of CorresNeRF with Reduced Correspondence Quantity

We obtained correspondences using image matching methods and subsequently sampled a subset (50%, 25%, 12.5%, 6.25%, and 3.125%) of these correspondences to train CorresNeRF. We then assessed CorresNeRF's performance on the LLFF dataset.

| Method                          | PSNR↑  | SSIM↑ | LPIPS↓ | Depth MAE↓ |
| ------------------------------- | ------ | ----- | ------ | ---------- |
| NeRF                            | 16.79  | 0.56  | 0.37   | 1.66       |
| CorresNeRF (with 3.125% corres) | 18.616 | 0.647 | 0.322  | 1.129      |
| CorresNeRF (with 6.25% corres)  | 18.934 | 0.657 | 0.299  | 1.113      |
| CorresNeRF (with 12.5% corres)  | 18.854 | 0.66  | 0.287  | 1.108      |
| CorresNeRF (with 25% corres)    | 19.068 | 0.669 | 0.269  | 1.10       |
| CorresNeRF (with 50% corres)    | 18.986 | 0.67  | 0.266  | 1.085      |
| CorresNeRF (with 100% corres)   | 19.83  | 0.70  | 0.29   | 0.91       |

Notably, even with only 3.125% of the correspondences, CorresNeRF significantly outperforms the baseline NeRF model. The performance of CorresNeRF enhances as the correspondence quantity increases. When 100% of the correspondences are used, CorresNeRF achieves its peak performance. Thus, as long as quality correspondences are provided in adequate numbers, CorresNeRF can surpass the regular NeRF's performance.

## Performance of CorresNeRF with more input views (3-view and 6-view)

We tested the robustness of CorresNeRF with varying input view counts. Specifically, we doubled the number of input views from 3 to 6 and then evaluated CorresNeRF's performance on the LLFF dataset.

| Method               | PSNR↑ | SSIM↑ | LPIPS↓ | Depth MAE↓ |
| -------------------- | ----- | ----- | ------ | ---------- |
| NeRF (3 views)       | 16.79 | 0.56  | 0.37   | 1.66       |
| CorresNeRF (3 views) | 19.83 | 0.7   | 0.29   | 0.91       |
| NeRF (6 views)       | 20.15 | 0.69  | 0.22   | 1.08       |
| CorresNeRF (6 views) | 21.51 | 0.74  | 0.22   | 0.85       |

The table indicates that CorresNeRF consistently outshines the baseline NeRF model, regardless of whether 3 or 6 views are used. Given that CorresNeRF is a plug-and-play module that can be added to any NeRF, provided quality image correspondences are available, its addition can boost the performance of NeRF, even in dense-view configurations.

## Using Different Image Matchers for CorresNeRF

We assessed CorresNeRF's performance using correspondences derived from different image matching techniques, namely LoFTR [55] and DKMv3 [56]. The LLFF dataset served as our testing ground.

| Method                  | PSNR↑ | SSIM↑ | LPIPS↓ | Depth MAE↓ |
| ----------------------- | ----- | ----- | ------ | ---------- |
| NeRF                    | 16.79 | 0.56  | 0.37   | 1.66       |
| CorresNeRF (with LoFTR) | 18.13 | 0.64  | 0.30   | 1.10       |
| CorresNeRF (with DKMv3) | 19.83 | 0.70  | 0.29   | 0.91       |

The results show that CorresNeRF can outperform the vanilla NeRF, irrespective of the image matching technique employed, as long as quality correspondences are acquired. Additionally, CorresNeRF benefits from a "free" performance boost when a superior image matching method is employed, offering avenues for further enhancing CorresNeRF's performance.

[55] LoFTR: Detector-Free Local Feature Matching with Transformers, CVPR 2022
[56] DKM: Dense Kernelized Feature Matching for Geometry Estimation, CVPR 2023"

---

> ### Author Response · Authors · 2023-08-14
>
> Dear Reviewer,
>
> We sincerely thank you for your precious time and efforts in reviewing our paper.
>
> We want to inquire whether our response has addressed your questions and concerns. We are more than happy to discuss with you further and provide additional materials.
>
> Best regards,
>
> Authors

---

> > ### Author Response · Authors · 2023-08-20
> >
> > Dear Reviewer,
> >
> > We sincerely thank you for your precious time and efforts in reviewing our paper.
> >
> > As we are approaching the deadline of the discussion period, we would like to inquire whether our response has addressed your questions and concerns. We are more than happy to discuss with you further and provide additional materials.
> >
> > Thank you again for the review and comments!
> >
> > Best regards,
> >
> > Authors

---

### Decision · Program_Chairs · 2023-09-21

**Decision:**

Accept (poster)

**Comment:**

The paper has some good ideas on the use of correspondences as priors, and achieves good results. The majority of the reviewers were positiveThe AC believes that the authors successfully rebutted the two negative reviewers.

It would be beneficial to include the CVPR 23 paper "SPARF: Neural Radiance Fields from Sparse and Noisy Poses" in related work, as the ideas are very similar, even though it was published after the submission.